# Multi-decadal increase of forest burned area in Australia is linked to climate change

Josep G. Canadell [1]✉, C. P. (Mick) Meyer[2], Garry D. Cook [3], Andrew Dowdy [4], Peter R. Briggs [1], Jürgen Knauer [1], Acacia Pepler [4] & Vanessa Haverd[1]

Fire activity in Australia is strongly affected by high inter-annual climate variability and extremes. Through changes in the climate, anthropogenic climate change has the potential to alter fire dynamics. Here we compile satellite (19 and 32 years) and ground-based (90 years) burned area datasets, climate and weather observations, and simulated fuel loads for Australian forests. Burned area in Australia's forests shows a linear positive annual trend but an exponential increase during autumn and winter. The mean number of years since the last fire has decreased consecutively in each of the past four decades, while the frequency of forest megafire years (>1 Mha burned) has markedly increased since 2000. The increase in forest burned area is consistent with increasingly more dangerous fire weather conditions, increased risk factors associated with pyroconvection, including fire-generated thunderstorms, and increased ignitions from dry lightning, all associated to varying degrees with anthropogenic climate change.

[1] Climate Science Centre, CSIRO Oceans and Atmosphere, Canberra, ACT 2601, Australia. [2] Climate Science Centre, CSIRO Oceans and Atmosphere, Aspendale, VIC 3195, Australia. [3] CSIRO Land and Water, CSIRO Land and Water, PMB 44, Winnellie, NT 0822, Australia. [4] Bureau of Meteorology, Climate Research Section, Bureau of Meteorology, Melbourne, VIC, Australia. ✉email: pep.canadell@csiro.au

The extraordinary forest fires in Australia in 2019 and 2020[1] have brought further interest in detecting changes in fire activity, the possible role of anthropogenic climate change and their likely future trends both in Australia and globally[2–6].

Terrestrial ecosystems in Australia are among the most fire prone in the world, with fire regimes varying widely[7,8]. Fire activity is dominated by savanna and rangeland fires in the northern and western parts of the continent characterized by fire return intervals of less than 5 years[7,9]. Forests in the east and south have fire return times of decades to more than a century, with subtropical and tropical forests in the northeast burning rarely or not at all[9]. Fire, including cultural burns by indigenous people, has shaped the function and structure of most Australian ecosystems for millennia[10,11].

Against this background of fire activity, Australia's mean temperature has increased by 1.4 °C since 1910 with a rapid increase in extreme heat events, while rainfall has declined in the southern and eastern regions of the continent, particularly during the cool half of the year[12–14]. These changes can affect the four components that must simultaneously come together for fire to occur: biomass production, its availability to burn (fuel loads), fire weather, and ignition[7], making Australian forests vulnerable and sensitive to changes in fire activity.

Previous studies showed increased fire danger due to changes in weather conditions over past decades in Australia[5,15,16], climate change fingerprinting to individual fire events and trends[17–19], and predicted increases in fire danger under future climate change due to increasing atmospheric concentrations of greenhouse gases[2,20,21]. Although these studies indicate more dangerous weather conditions for wildfires in a warmer world, studies also suggest that trends due to climate change might not be clearly detectable until later in the coming decades owing to the high natural variability and extremes of the Australian climate[4,22–24].

Fuel loads and trends, as effected by climate, human activity and time since the last disturbance, also play a role in determining fire risk[25,26]. This link is a central motivation for using prescribed burning to reduce fuel availability[27], which in Australia is managed through changes in the frequency of prescribed burns[28]. Although there is some debate on their value to reduce fire risk[29], particularly during extreme fire weather conditions[2,30], fuel loads and their distribution and structure are key determinants of fire spread, intensity and severity[7].

Here we analyze trends of the burned area in forest ecosystems in Australia, which are dominated by temperate forests extending over the southern and eastern regions of the continent. We use a high-resolution (1.1 km x 1.1 km) burned area satellite record available based on NOAA-AVHRR (32 years), the NASA-MODIS burned area at 500 m resolution (19 years), and the fire histories from State and Territory government agencies (90 years). In addition, we analyze trends of nine wildfire risk factors and indices that relate to characteristics of fuel loads, fire weather, extreme fire behaviour, and ignition, which together with the burned area enable us to infer the causal influence of climate change on fire activity.

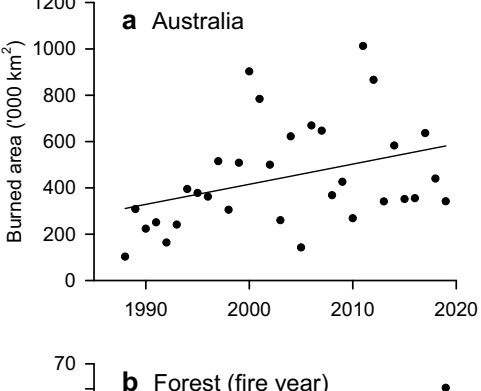

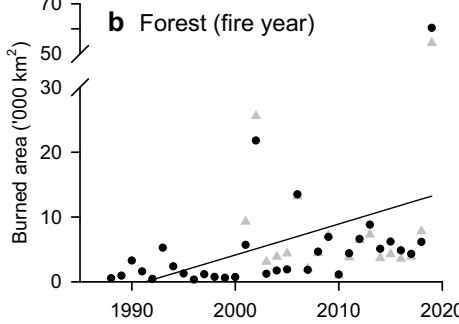

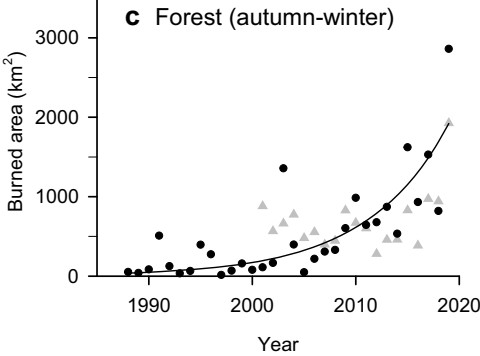

**Fig. 1 Burned area (km²) for each fire year (July to June) from 1988 to 2019. a** Whole of continental Australia including Tasmania, linear fit; (**b**) Australian forests, linear fit; and (**c**) Australian forests for the Austral autumn/winter seasons (March to August), exponential fit (regressions in Supplementary Table 1). Data: AVHRR-Landgate (1988–2019, dots) and MODIS (2002–2019, triangles). Regressions are calculated using AVHRR-Landgate, and MODIS is shown for comparison.

## Results

**Trends in area burned.** At a continental scale, total annual burned area (fire year defined as July to June to include the Austral summer of December to February) using the NOAA-AVHRR dataset ("Methods": Burned area data), significantly increased over the past 32 years albeit with large interannual variability (Fig. 1a; Linear fit, $p$ value = 0.04, Supplementary Table 1). The high variability is in part driven by large-scale modes of atmospheric and oceanic variability such as El Niño Southern Oscillation (ENSO) and the Southern Annual Mode[31,32] that influence fire weather conditions[16,22]. Nine out of the 11 fire years, each with more than 500,000 km² (>50 Mha) burned, occurred since 2000.

Forest ecosystems also show increased burned area over time (Fig. 1b, linear fit, $p$ value = 0.02, Supplementary Table 1; Fig. 2). The increasing trend is statistically significant with and without the 2019 fire year, indicating a robust increasing trend even before the extraordinary large burned area of that year (Supplementary Table 1). Forests in Australia experienced an annual average increase of 350% in burned area between the first (1988-2001) and second (2002-2018) half of the record, and an increase of 800% when including 2019. The 2019 fire year burned about three times (60,345 km²) the area of any previous year in the 32-year AVHRR-Landgate record (Fig. 3, Supplementary Fig. 1, "Methods": Burned area). The burned area of the 2019 fire year was estimated at 71,772 km² based on State and Territory agencies (NIAFED) and 54,852 km² based on NASA-MODIS,

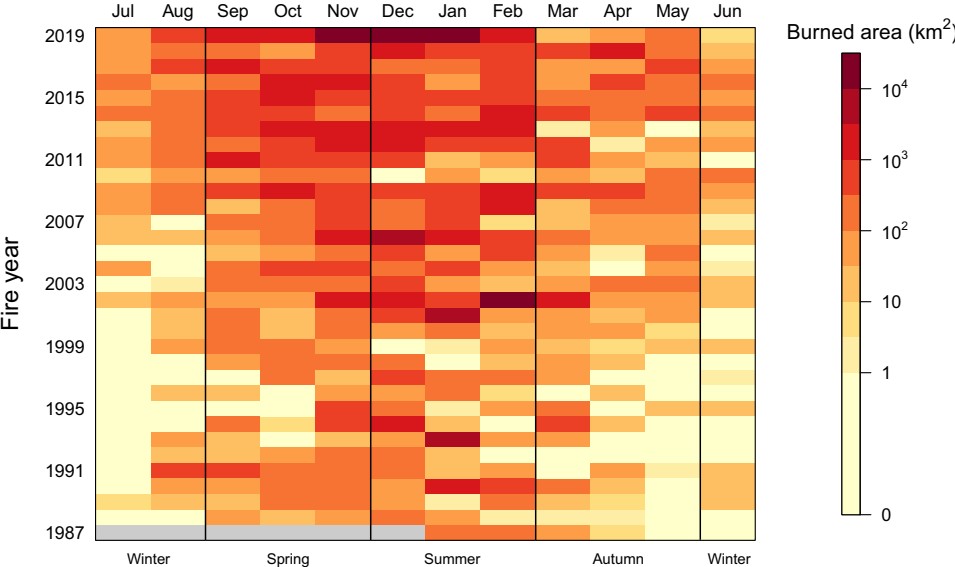

**Fig. 2 Monthly burned forest area for fire years (July to June).** Data: AVHRR-Landgate (1988–2019).

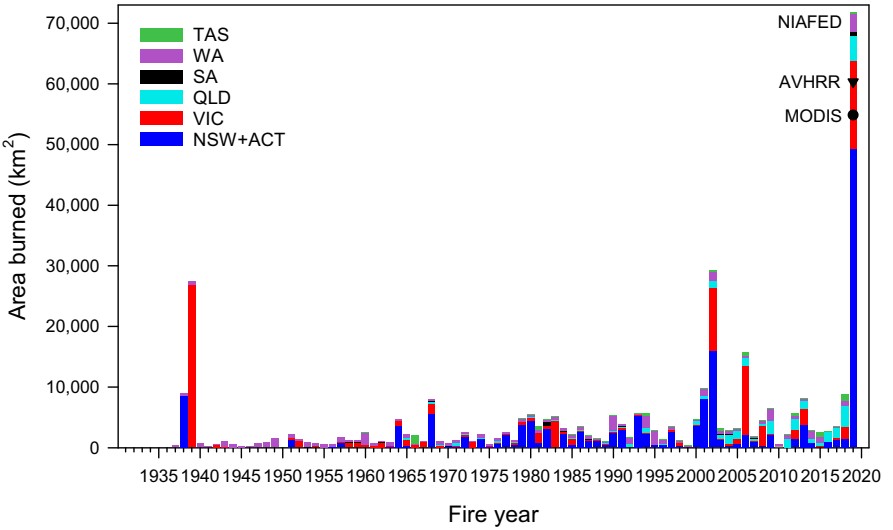

**Fig. 3 Wildfire burned area by states and territory in forest ecosystems for the 1930 to 2019 fire years.** New South Wales and Australian Capital Territory (dark blue), Victoria (red), Queensland (light blue), South Australia (black), Western Australia (violet) and Tasmania (green). Data for 1930–2018 stacked bars are State and Territory agencies fire histories, supplemented with MODIS for Queensland in 2016–2018. Data for 2019 fire year are National Indicative Aggregated Fire Extent Dataset (NIAFED) by States and Territories (stacked bar), AVRHRR-Landgate (filled triangle), and MODIS (filled circle). See "Methods": Burned area data.

with an average for the three products of 62,323 ± 8,631. Ten out of eleven fire years with at least 5000 km$^2$ (>0.5 Mha) burned have occurred since 2001. These trends are broadly consistent across the three burned area products (Supplementary Fig. 1).

We find a positive exponential trend of burned area in forest ecosystems during the cool season months of the Austral autumn and winter (March to August), with a mean growth rate of 14% yr$^{-1}$ (Fig. 1c; exponential fit, p value = 0.001; 9% increase without the 2019 fire season; Supplementary Table 1). These results indicate an extension of the fire season into the cooler months of the year, with more than a five-fold increase in the annual mean burned area in winter and three-fold increase in autumn between the first and second half of the studied period. However, spring and summer contributed about 10 times more to the increase in burned area than autumn and winter (Fig. 2; Supplementary Figs. 2 and 3). All seasonal linear trends were highly significant (p

value < 0.001), and with p value = 0.07 for the summer season. The seasonal fractions of burned area for the 32-year period are 66% in summer, 24% in spring, 7% in autumn and 3% in winter.

Along the latitudinal increasing temperature gradient from South to North (Tasmania, Victoria, New South Wales, and Queensland), we find that the largest relative growth of burned area between the early (1988–2002) and later (2003-2019) periods of the record occurred in the southern coolest parts of the forest distribution (Tasmania) and the northern warmest parts of the forest distribution (Queensland), with Queensland showing the largest absolute difference of the two.

The AVHRR-Landgate data predominantly detects wildfires and misses low-intensity fires, including most prescribed (planned) burning[33], suggesting the trends presented here are very unlikely to be affected by changes in prescribed burning ("Methods": Burned area data). This result is further corroborated

by the lack of trends in the annual area of prescribed burning (see section below Fire risk factors and fuel load).

Fire histories compiled from State and Territory agencies ("Methods": Burned area data), suggest that the extent of burned area for the 2019 fire year was also unprecedented since the 1930s when most agencies began to collect annual records (Fig. 3). Out of four forest megafire years (defined as the top 10 percentile years with most burned area) that have occurred since 1930, each with over 10,000 km² (1 Mha) of burned area (1930, 2002, 2006, 2019), three occurred since 2000. Historical records show that no other forest megafire years occurred during the 1900s but that a large fire year occurred in Victoria in 1851[34]. Further disaggregation into States and Territories, and wildfires and prescribed burning is shown in Supplementary Fig. 4.

The increasing trend in annual burned area, and the exceptional 2019 fire year, become all the more significant

against a concurrent diminishing of forest extent available to burn due to land clearing for pasture and agriculture[35], and increasing bushfire firefighting capacity in Australia[36,37].

**Years since the last fire.** We further investigate trends in increasing burned area with an analysis of the number of years since the last wildfire (YSLF). The longer time series of the fire history records from State and Territory agencies enable us to construct a gridded database of wildfires at 250 m resolution to analyze decadal changes in the number of YSLF for all pixels that have burned at least once ("Methods": Years since the last fire). The analysis shows that 48.8% of all forest area has burned at least once since the 1930s. The burned fraction of the total area for each of the forest classes consistent with the Australian National Forest Inventory ("Methods": Forest extent and types) varied widely: eucalypt low-forest 61.1%, eucalypt tall-forest 60.6%, eucalypt medium-forest 58.2%, coastal non-eucalypt 54.3%, and rainforest 11.1%. Thus, YSLF estimates presented here do not represent a fire frequency for specific region or forest type, but they are correlated, as they have been estimated by a combination of observations and expert knowledge elsewhere[9].

For context, some of the dominant vegetation types such as tall eucalypt forests have typical fire intervals of 20 to 100 years with extreme cases with more than 100 years[9]. Lower stature eucalypt forests have typical fire intervals between 5 and 20 years with extreme cases with a range between 20 and 100 years[9]. Tropical forests have fire return intervals above 100 years[9].

The YSLF declined over the past four decades with decadal means (±SD) (all forests that had burned at least once since 1930s) of 70.6 ± 1.1 years, 68.1 ± 1.0 years, 53.2 ± 7.8 years, and 39.8 ± 8.1 years for the decades of 1980s, 1990s, 2000s, and 2010s, respectively (Fig. 4; continental figures in Supplementary Fig. 5). YSLF values for the last four decades of the record are the most robust due to the large amount of data from which they are estimated (cumulative), and therefore are the focus of this analysis. Fig. 4 reveals areas with rapidly declining YSLF, with some regions in Victoria with YSLF below 20 years. At the State and Territory level, the decade of 2010s

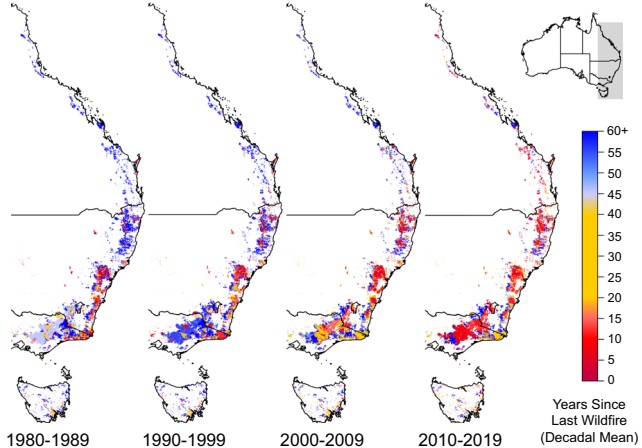

**Fig. 4 Number of years since the last wildfire (decadal mean) for forested areas.** Analysis based on forested areas that have burned at least once since fire records began in the 1930s for most states. Spatial resolution is 250-metres. Data: State and Territory fire histories.

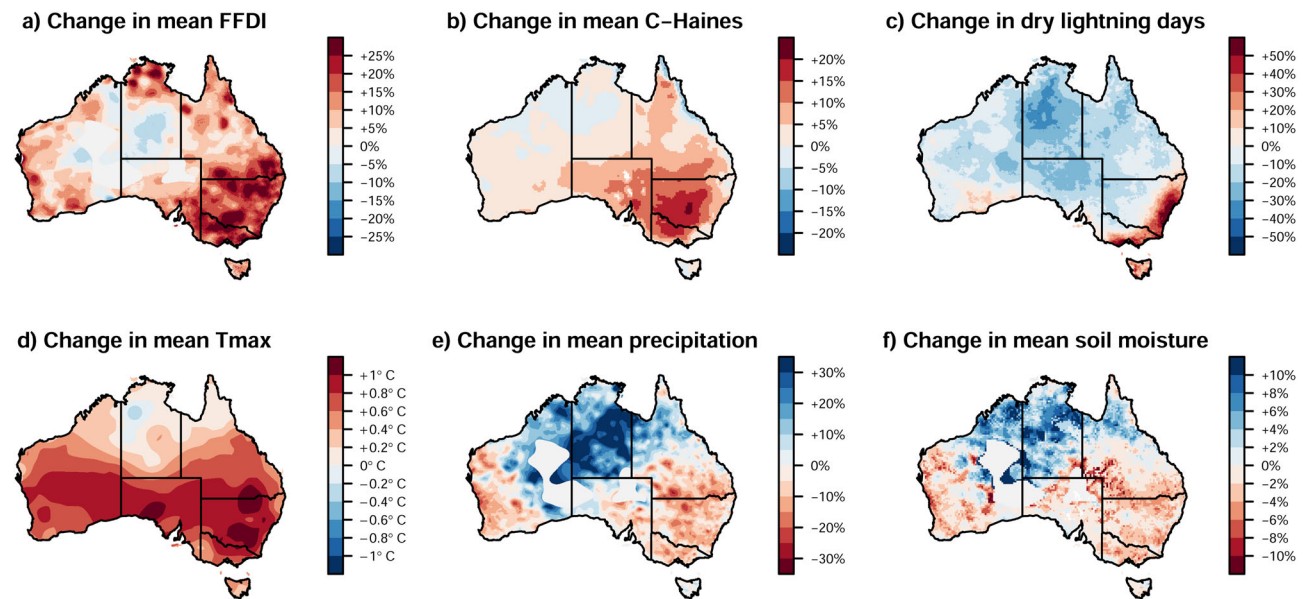

**Fig. 5 Climate change factors associated with wildfire weather and activity. a** Near-surface fire weather conditions based on the Forest Fire Danger Index, (**b**) mid-tropospheric fire weather conditions based on the C-Haines Index, (**c**) dry lightning conditions as key factors for ignitions, (**d**) daily maximum temperature, (**e**) annual rainfall deficit and (**f**) soil moisture (0–23 cm) associated with dryness of the forest system. Changes are calculated as the changes from 1980–1999 to 2000–2019, calendar years, for all variables except for dry lightning to 2000–2016. Grey areas in (**e**) and (**f**) denote areas with insufficient data availability.

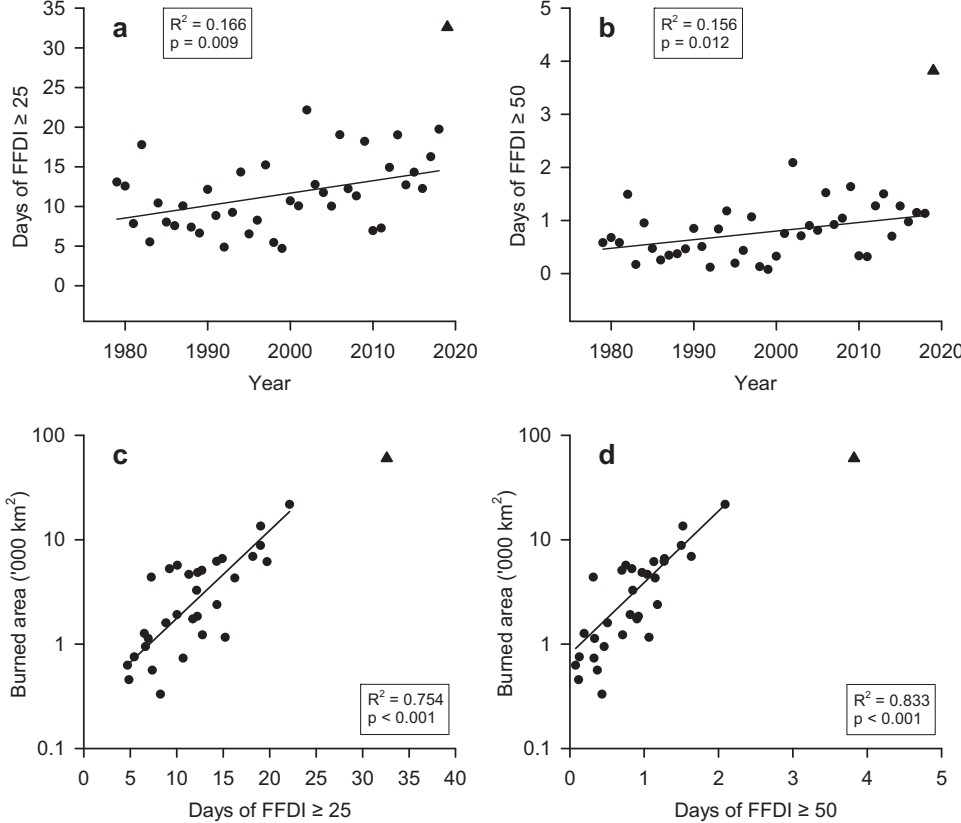

**Fig. 6 Trends in the number of days with very high or severe Forest Fire Danger Index.** Trends in the number of days in which Forest Fire Danger Index (FFDI) equals or exceeds (**a**) 25, linear fit, or (**b**) 50, linear fit, over the fire years of 1979 to 2018 and averaged over forest ecosystems in Australia (Supplementary Table 1). Relationship between FFDI (**c**) ≥ 25 (exponential fit) and (**d**) ≥50 (exponential fit) and burned area for the fire years of 1988 to 2019 (Supplementary Table 1, Supplementary Fig. 6c,d). 2019 fire year (triangle). Burned area data: NOAA-Landgate (1988–2019).

shows a range from $46 \pm 21$ in Queensland to $33 \pm 4$ in Victoria, with a minimum of $18 \pm 4$ years for the Australian Capital Territory whose forests were largely burned in 2003 and again in 2020 (Supplementary Table 2).

**Fire risk factors and links to climate change.** There are four components that drive fire activity: weather that influences fire spread, ignition sources, biomass accumulation (fuel loads), and dryness of fuel loads (fuel availability)[2,7]. In the next three sections we address each of these four factors and show trends of various indices and quantities which we link back to changes in fire risk over the past 40 years.

We use a daily dataset of McArthur Forest Fire Danger Index (FFDI) values gridded throughout Australia[15]. This index is specifically developed to indicate weather conditions across the "Danger" spectrum associated with wildfires in Australian forests[38]. It integrates a combination of key weather factors known to influence the severity of wildfires including wind speed, humidity, temperature, and fuel moisture deficit, which is calculated from antecedent rainfall and temperature.

Multi-decadal changes towards more dangerous fire weather (as indicated by FFDI) are occurring for most of Australia (Fig. 5a). The largest increases occur in the southeast, including in association with increased temperatures (Fig. 5d) and decreased rainfall (Fig. 5e) in that region, as others have reported before[2,5,15,39]. Temperature influences fire behaviour in a number of ways. First, through relative humidity (vapour pressure deficit) and its effect on the absorption of energy by the atmosphere and on the moisture content of vegetation. Second, by influencing atmospheric stability based on temperature lapse rates and humidity profiles, which influences the risk of

extreme wildfire events including those that generate new thunderstorms[20,21,40]. Third, through its contribution to the occurrence of heat waves[18], and fourth by influencing the speed and intensity of drought development[41,42], along with other factors such as wind speed, specific humidity and soil moisture.

Over the last four decades, a trend has been observed towards drier conditions during the cooler months of the year (e.g., April to October) in many parts of southeastern Australia (Fig. 5e for annual change). This has been associated with a strengthening of the subtropical ridge and a decrease in rainfall from fronts and cyclones[13,14,43]. The observed trends are consistent with projected future declines in cool season rainfall in the 21st century from global climate models, and are stronger than those produced in historical simulations for the period 1950-2005[44,45]. Reduced Austral spring and winter rainfall influences the fuel moisture leading into the fire season, with drier fuels in spring being associated with a trend towards an earlier start to the fire season particularly in southeast Australia in recent decades[15,16,24].

In addition to the FFDI, which is based on weather conditions near the surface, we used the Continuous Haines (C-Haines) index[46] to represent dangerous weather conditions for wildfires based on higher levels of the troposphere ("Methods": Fire weather indices and climate/weather variables). The C-Haines index is used to indicate a high chance of extreme wildfires such as those with thunderstorm formation in fire plumes (generally referred to as pyrocumulonimbus cloud, termed pyroCb)[21,40,46]. PyroCbs are associated with extreme dangerous fire behaviour, including erratic and strong gusty winds near the surface and generating lightning in the fire plume that can ignite additional fires far ahead of the main fire front[47,48].

Long-term changes in the C-Haines Index have occurred over the last four decades in many parts of Australia (Fig. 5b), with higher values representing more dangerous conditions for extreme wildfire events, particularly over forest ecosystems of southeast Australia. This is consistent with previous studies that have demonstrated significant increases in C-Haines Index in recent decades, particularly for regions of southern Australia[40]. High C-Haines index occurred in previous wildfire disasters in southeast Australia including during the Black Summer fires in 2019/2020, the Black Saturday fires in Victoria in 2009, and the Canberra fires in 2003[40,49]. Often high C-Haines values are accompanied by high FFDI values (Supplementary Table 3), but there are exceptions with a notable case for the Canberra fires in 2003 where the C-Haines had a higher percentile value than FFDI, although both were high[40].

Modelling shows that increasing anthropogenic greenhouse gas emissions can increase the C-Haines in some regions including southeastern Australia[20,21]. These climatological changes are primarily associated with an increase in the moisture-related component of the C-Haines (representing increased dewpoint depression at 850 hPa) rather than its stability component as seen based on historical reanalysis data[40] and projections of future climate[20]. Increased dewpoint depression (and similarly relative humidity or vapour pressure deficit) is primarily caused by increased temperatures, but countered to some degree by an increase in atmospheric moisture content that is as expected consequence of a warmer world[50].

Fire ignition factors in Australia include multiple anthropogenic and natural sources. The primary natural cause is lightning, particularly that known as "dry lightning", which occurs when the lightning is not accompanied by substantial rainfall. A threshold of rainfall of about 2.5 mm is used to define dry lightning. Rainfall less than this quantity on a day with lightning is associated with a higher than average chance that the lightning will result in a wildfire[51]. Lightning can be responsible for more than 50% of the area burned in some regions of Australia including the temperate forest regions of southeast Australia[52]. Although lightning observations are not available in a homogenous form over many decades, environments associated with dry lightning events show an increased frequency of occurrence in recent decades in near-coastal regions of southeast Australia (Fig. 5c)[53]. Specifically, there is an increase of about 50% in the frequency of occurrence of those environments for the recent period of 2000–2016 compared with the previous two decades of 1980–1999.

**Burned area and fire weather**. To understand the influence of fire weather on the observed trends in burned areas in forest ecosystems, we use uni- and multi-variate regression analyses with the fire risks factors of FFDI ≥ 25, FFDI ≥ 50, C-Haines and dry lightning as predictor variables. Note that we find no significant relationships with biomass variables (see section Fire risk factors and fuel loads, and Fig. S7), and therefore they are not included in this analysis.

We found both FFDI equal or exceeding 25 (Very High fire danger classification starts at FFDI ≥ 25) and equal or exceeding 50 (Severe fire danger classification starts at FFDI ≥ 50) to be significantly increasing over the past four decades for the forest areas (Fig. 6a,b; linear fit, $p$ value = 0.01, Supplementary Table 1). The five years with most severe or greater fire danger (FFDI ≥ 50) all occurred since 2002 (2002, 2006, 2009, 2013, 2019).

For the last 32-year period (1988–2018, section Trends in burned area), burned area in forest ecosystems increased exponentially with the number of days in which FFDI was equal to or greater than either 25 or 50 (Fig. 6c,d, exponential fit, $p$ values < 0.001; Supplementary Table 1; Supplementary Fig. 6 for

exponential fits). These trends equate to 21% increase in the burned area for every additional day of FFDI ≥ 25, and about 3 to 5 times increase in the burned area for every additional day of FFDI ≥ 50.

Using a multivariate regression analysis, we show that FFDI is a strong predictor of burned area for both FFDI ≥ 25 and FFDI ≥ 50, as FFDI alone is able to explain more than two-thirds of the variance in both models (Supplementary Table 3). Including C-Haines and dry lighting into the regression model improves the explained variance only marginally, most likely due to fact that these two variables are highly correlated with FFDI (Supplementary Fig. 7).

We use the exponential regression of burned area against days of very high FFDI ≥ 25 for the period 1988–2018 (Supplementary Table 1, regression no.8) to test whether it could have predicted the record burned area of the exceptional 2019 fire year (June 2019 to July 2020). The regression model predicts a total area burned of 143,151 km² against the three estimates available (71,772 km² Agencies-NIAFED, 60,345 km² AVHRR-Landgate, 54,852 km² NASA-MODIS) ("Methods": Burned area data). The regression model was not developed as a predictive tool for burned area under future climate conditions, however, it clearly predicted a record megafire year for forests for the 2019 fire year, as occurred. These results suggest the very likely possibility of further increases in burned area in response to the predicted higher FFDI values under future warming scenarios[2,6,20,21,54].

**Fire risk factors and fuel load**. In addition to weather and climatic conditions, and the availability of ignition sources for a fire to occur, fuel amount, structure, continuity and condition are also key components of fire risk[26,29]. The availability of fuel plays a key role in determining the intensity (peak energy output) and severity (impacts related to the amount of canopy scorch, canopy loss, tree death, biomass loss) of fires. Understanding the role of fuel loads in fire activity is important as it also provides the knowledge base to reduce fire risk through the management of fuel quantity and distribution in the landscape[26,30]. Fuel loads are composed of aboveground biomass (particularly leaf biomass) and litter, including coarse woody debris (e.g., dead branches, logs, standing dead trees) and fine litter (eg, leaf, twigs).

There is no continental forest observatory available that tracks fuel loads which could enable a similar analysis to the one above for climate/weather factors and burned area. However, regression analyses with modelled fuel loads show no statistically significant relationships with burned area (Fig. S7). This result does not rule out the possible role of fuels in influencing burned area, but it is the reason for not including fuels in the multi-variate analysis above.

Here we analyze two datasets that characterize processes influencing fuel loads and therefore fire risk: (1) changes in the area burned by hazard reduction fires; and (2) the influence of climate change and increasing atmospheric $CO_2$ concentrations on biomass production and fuel loads. We hypothesize that changes in trends leading to reduced fuel loads will reduce fire risk and burned area, while the opposite will hold true for changes leading to increased fuel loads.

First, we address hazard reduction burns, also known as prescribed fires, planned fires or fuel reduction burns, including cultural burns by indigenous people. These fires are deliberately human-lit during the cooler months of the year leading to reduced fuel amounts. They have multiple purposes including the prevention or reduction of damage to human life and infrastructure, biodiversity conservation, reducing future fire intensity and increasing hunting for First Nations communities. We extracted the prescribed burning areas in forest ecosystems

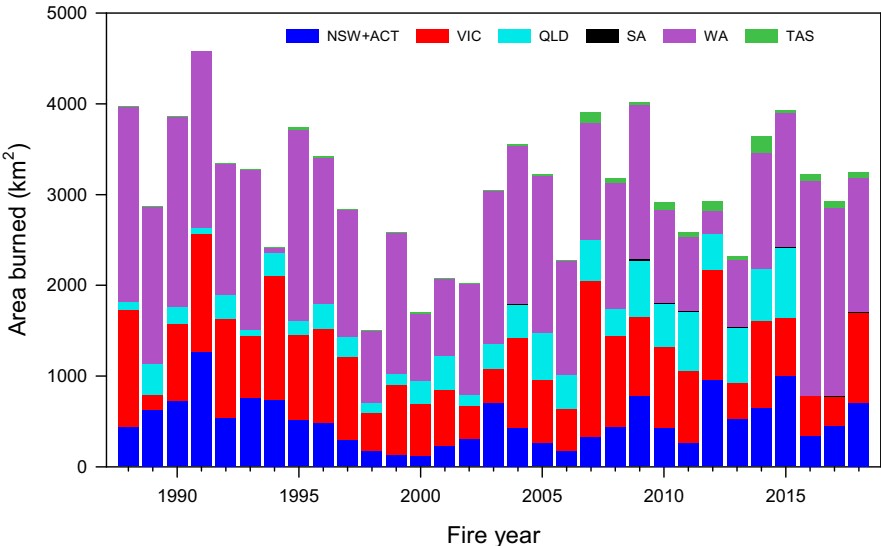

**Fig. 7 Area of prescribed burning in forest ecosystems.** Data from the States and Territories fire histories for New South Wales and Australian Capital Territory (NSW + ACT, dark blue), Victoria (VIC, red), Queensland (QLD, light blue), South Australia (SA, black), Western Australia (WA, violet) and Tasmania (TAS, green). Queensland data for 2016-2019 is not available.

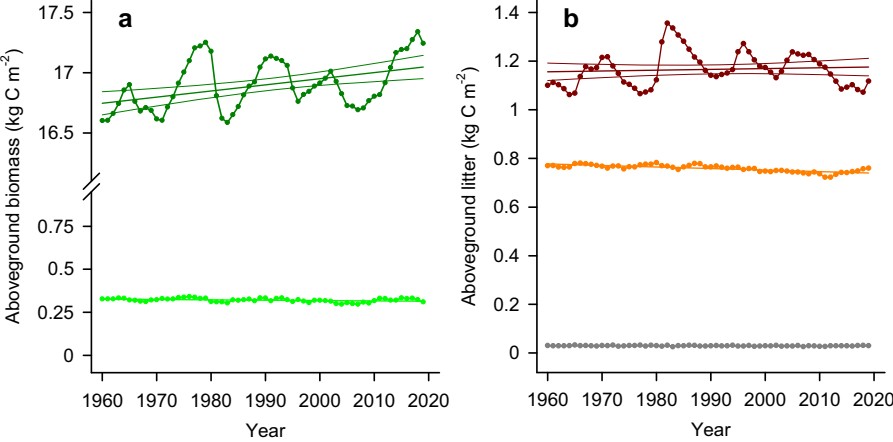

**Fig. 8 Modelled aboveground biomass and litter fractions of forested areas. a** Aboveground woody biomass (dark green, with 95% confidence interval) and leaf biomass (light green), and (**b**) litter fractions of forests derived from BIOS-CABLE with varying observed $CO_2$ and climate. Coarse woody debris (dark red, with 95% confidence interval), fine litter (orange), very fine litter (grey).

over the past 32 years from the State and Territory fire history databases ("Methods": Burned area data). Prescribed fires burned an average of 3071 ± 732 km² per year, or about 1% of the current area of forest ecosystems. There is large inter-annual variability due to the year-to-year variability in suitable weather conditions to conduct prescribed burns, but the data show no trends over the past three decades (1988–2018) in forest ecosystems (Fig. 7).

Second, we address the influence of climate change and the associated accumulation of anthropogenic $CO_2$ emissions in the atmosphere on biomass production and fuel loads through changes in temperature, rainfall and the elevated $CO_2$ effect on vegetation growth (i.e., the $CO_2$ fertilization effect)[55,56].

Here we use a highly parameterized and benchmarked biospheric model for Australia (CABLE)[57–59] for the estimation of fuel types and loads[60]. CABLE was forced with observed $CO_2$ concentration and climate over the past four decades, but does not include the effects of fire and other disturbances, hence we refer to the simulated rates as the potential rates of fuel production in the absence of fires. The simulation shows a small positive trend of potential aboveground biomass and coarse

woody debris (Fig. 8a, b; Supplementary Table 4), but a declining trend of fine litter associated with declining canopy leaf biomass over Australian forests (Fig. 8b). This declining trend could be associated with the reduction in rainfall and soil moisture of the past four decades (Fig. 5e, f). In addition to declining fine litter and its associated fire risk, we find a decrease of rainfall and soil moisture down to 23 cm depth over the past 40 years, as seen in the historical record (Fig. 5e) and simulated by CABLE-BIOS as a proxy for reduced water availability at the forest floor (Fig. 5f) (see "Methods": Biomass and fuel loads). This suggests a likely increase in dryness of fine litter, which is associated with increases fire risk[61]. Fine fuels are a key determinant of fire risk and initial spread[25], while coarse fuels are more associated with total energy output and overall fire severity[26].

We isolate the effects of the $CO_2$ fertilization from those of climate change (e.g., changes in temperature, rainfall, wind) on the observed biomass and fuel trends with an additional simulation ("Methods": Biomass and fuel load). The elevated $CO_2$ effect led to an increase in potential aboveground vegetation (biomass and litter) of 11% during the period 1960–2018 but with

an overall net increase of only 9.5% due to the offsetting effects of changing climate (Fig. 9).

## Discussion

We find an annual linear increase in the forest area burned in Australia over the past 32 years, with increases in all seasons and the largest interannual variability during summer. The growth in burned area is exponential for the autumn-winter period, although the largest absolute seasonal contributions to the increase were in spring and summer. These trends show a lengthening of the fire season towards the cooler seasons, where limited fire activity occurred in the early part of the satellite 32-year record. We also find that along the latitudinal temperature gradient South to North, fire activity increased the most at both ends of the gradient, with fires increasingly intruding into fire-sensitive forests: alpine forests in Tasmania in the South and tropical rainforests in Queensland in the North. Both trends, spatial and temporal, provide robust evidence for an increase in fire activity across forest ecosystems in Australia.

In addition, we find that the frequency of megafire years in forests with more than 10,000 km$^2$ (1 Mha) burned has markedly increased since 2000. Together these trends have led to the rapid decrease in the number of years since the last fire. The mean YSLF values across the last four decades are still within the historical fire regimes reported for the same type of forests[9], however, we find a growing trend in the number of pixels in which YSLF is decreasing (Fig. 4). A continuation of that trend, especially in areas with sensitive species, could lead to significant ecological changes. Examples of sensitive ecosystems are the mountain ash (*Eucalyptus regnans*) and alpine ash (*Eucalyptus delegatensis*) forests in Victoria. Both species are obligate seeders that require 20-30 years to reach maturity and begin to establish a seedbank able to recover the forest after fire[62–64]. Some regions in Fig. 4 show mean YSLF of about 20 years or less for the last decade. At this short fire return interval, if fires were to be of high severity leading to tree mortality (which this study has not assessed), both ecosystem-types would be at risk of local collapse.

We also show an increasing trend in very dangerous fire weather based on the FFDI for near surface conditions and C-Haines for lower to mid-tropospheric vertical atmospheric stability and humidity measures relevant to pyroconvective processes, consistent with previous studies[15,40]. We find strong correlations between those fire weather indices, long-term changes in climate and dry lightning with increase forest burned area. These multiple and individual correlations establish a robust and multi-evidence link between climate change and increased fire activity over the past three decades. Both increased temperature and shifts in large-scale rainfall patterns have very likely played a dominant role in observed burned area trends.

Two independent analyses were performed to investigate the possible role of prescribed burning and climate change on fuel loads. We found no changes in the mean annual area of prescribed burning over the past 32 years, although we have no information on how successful those burns were in reducing fuel loads. However, given the lack of trend and the fact that on average, only 1% of forests are subject to fuel reduction burns every year, it is very likely that fuel management had no effect on the observed multi-decadal increasing trend in the burned area of forest fires. We also note that the main objective of fuel management is to reduce fire risk and severity[27], which might or might not result in reduced total burned area.

The evidence for the possible role of $CO_2$ and climate effects on fuel loads is complex, with increased modelled total biomass due to elevated $CO_2$ partially offset by the negative effects of climate change. The actual and future effects as climate and atmospheric

$CO_2$ continue to change is uncertain, with the only elevated $CO_2$ experiment on eucalyptus forests in Australia finding no effect of $CO_2$ on biomass production due to nutrient limitation[65]. We also found a potential decline in fine fuels, but increased dryness as suggested by proxy data, which led to decreased and increased fire risk, respectively.

Together, these results further underscore the dominant role of changes in fire weather and climate on the observed trends in burned area, and the need to further investigate the role of fuel loads in a changing environment.

Based on the strong correlations between FFDI and the observed trends of a number of other factors associated with climate change and fire weather, our analyses suggest that the observed trend of increasing burned area and frequency of forest megafires are likely to continue under future projected climate change. Consequently, there are considerable implications for emergency management, health, infrastructure, natural resource management and ecological conservation, noting the severe impacts that the 2019/2020 fire season had on these sectors in Australia.

## Methods

**Forest extent and types**. The forest region in this study is based on the agro-ecological zones of the Biogeographic Regionalization for Australia (IBRA) (https://www.environment.gov.au/land/nrs/science/ibra), with the corresponding forest types from the National Forest Inventory[66]. Forest ecosystems are dominated by open forests of up to 80% canopy cover and 30 metres height and woodland forests up to 50% canopy cover. The forest vegetation classes were developed from ground, aerial and remotely sensed vegetation mapping, the National Vegetation Information System (NVIS), and large field validation programs (https://www.environment.gov.au/land/nrs/science/ibra,[66]). The following vegetation types are included: Eucalypt low closed, Eucalypt low open, Eucalypt low woodland, Eucalypt medium closed, Eucalypt medium open, Eucalypt medium woodland, Eucalypt tall closed, Eucalypt tall open, Eucalypt tall woodlands, Rainforest, Leptospermum, Banksia, Other native forest, Softwood plantation, Hardwood plantation, Mixed species plantation and Other forests (unallocated types). ArcGIS majority resampling was used to determine forest/non-forest membership of cells at 0.01° resolution (1.1 km × 1.1 km) for comparison with Landgate NOAA-

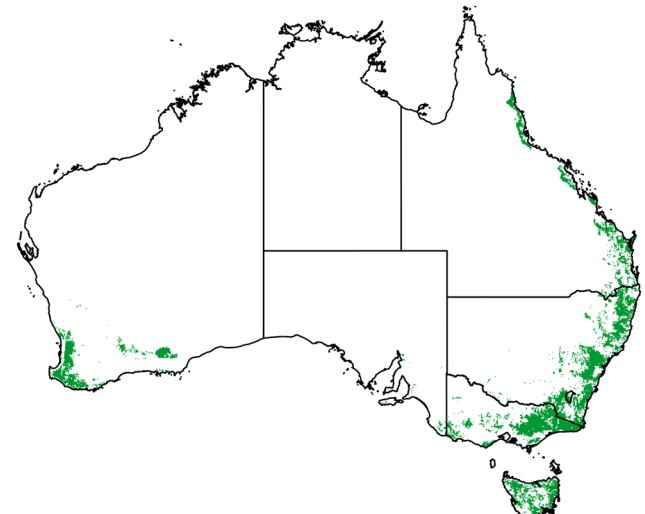

**Fig. 9 Forest ecosystems distribution used in our study.** The following vegetation types from the Australian National Forest Inventory are included (250 m spatial resolution): Eucalypt low closed, Eucalypt low open, Eucalypt low woodland, Eucalypt medium closed, Eucalypt medium open, Eucalypt medium woodland, Eucalypt tall closed, Eucalypt tall open, Eucalypt tall woodlands, Rainforest, Leptospermum, Banksia, Other native forest, Softwood plantation, Hardwood plantation, Mixed species plantation and Other forests (unallocated types). Nominal forest types occurring in savanna, rangeland, and littoral ecosystems are not included: Callitris, Casuarina, Eucalypt Mallee, Mangrove, Melaleuca.

AVHRR, and at 0.0025° (~250 m × 250 m) for all other analyses. The total area covered by this study is 324,873 km$^2$.

### Burned area data

*A national satellite-based burned area (Landgate NOAA-AVHRR).* We use a database of burned area developed for Australia covering the whole continent for the period 1988 to June 2020. The data product is developed by the Western Australia Land Authority (Landgate) that produces a burned area based on NOAA's Advanced Very High-Resolution Radiometer (AVHRR) at 1.1 km x 1.1 km and monthly resolution with continental coverage (https://myfirewatch.landgate.wa.gov.au). The product has been validated with fire history data from State and Territory government agencies, higher resolution products (eg, MODIS) and field observations[67,68] including for forests in southeastern Australia[33]. It is currently used for web-based fire mapping and applications (https://myfirewatch.landgate.wa.gov.au) and greenhouse emissions accounting for the whole of Australia[69,70]. These data were resampled from the ESRI format shapefiles supplied by Landgate to 0.01° rasters using the GDA94 datum.

We use July-to-June fire years in all burned area data analyses to include the full Austral spring and summer (September to February) when most forest wildfires occur, including the 2019 fire year.

The NOAA-Landgate burned area has been shown to have up to 87% omission rate for prescribed burning in southern temperate forests[33]. Prescribed burning is of low intensity and sub-canopy fires that are largely concealed from satellite detection. For the objectives of our analyses, the spatial and temporal consistency of the NOAA-AVHRR data is the most important feature to reveal changes in long-term trends, and the omission of most prescribed burns is desirable to remove the direct influence of human management. This approach enables assessment of the links between wildfires and fire weather and climate. NOAA-AVHRR burned area is also likely to miss low intensity wildfires. The data do not have attributes on the intensity and severity of the fires.

*State and territory fire histories data.* We collected burned areas from land management and fire suppression agencies from all States and Territories in Australia. Agency data are provided as polygons in vector files derived from a variety of sources. Data include paper records, ground mapping, and aerial photography for the earlier part of the record, and GPS boundary tracing from ground surveys, line scans, aerial photography and, in some cases, satellite mapping for the later part of the record. These are converted to rasters at 0.0025 deg resolution.

The responsibility for fire mapping, firefighting and management, including prescribed burning, falls within the jurisdictions of States and Territories, which means that there is no nationally-consistent approach, that methodology has changed over time, and there is not a centralized database[71]. The records discriminate wildfires from prescribed burning, and most agencies began annual fire recordings by the 1930s: New South Wales (1900), Victoria (1900), Western Australia (1922), Queensland (1930), South Australia (1932), Tasmania (1960), with data quality increasing over time (see data sources in Supplementary Table 5). The data have been extensively validated, particularly over the past few decades, and researched (e.g.,[2,17,29,61,64]). The data do not provide information on the severity of the wildfires or how successful prescribed burns were in reducing fuel loads.

Data for the 2019 fire year (July 2019 to June 2020) came from the rapid-response National Indicative Aggregated Fire Extent Dataset (NIAFED) developed collectively by the same State and Territory agencies. The NIAFED v20200622 used a variety of mapping methods (including ground and aerial surveys, and remote sensing) and attribution approaches. The fast nature of the assessment means the dataset lacks national coherency, and tends to be an overestimate due to its precautionary nature. Accessed 13 October 2020 http://www.environment.gov.au/fed/catalog/search/resource/details.page?uuid={9ACDCB09-0364-4FE8-9459-2A56C792C743}.

The last complete annual data for the state of Queensland were for 2015 and were completed to June 2020 with the NASA-MODIS burned area product (see NASA-MODIS burned area below). We chose the MODIS product, and not the NOAA-Landgate burned area, to have two totally independent products (Agency histories and NOAA-Landgate) for long term-trend analysis.

*NASA-MODIS burned area.* NASA-MODIS burned area data were also used as a third estimate for the 2019 fire year and to fill gaps in recent data for the state of Queensland (see previous section). Its shorter coverage (18 years) makes it less valuable to study trends.

The NASA-MODIS burned area, at monthly and 500 m resolution, combines MODIS Thermal Anomalies and the Fire product MCD64A1.006[72]. The data were downloaded for the Australian region in cylindrical equidistant form (0.004167° resolution) at https://lpdaacsvc.cr.usgs.gov/appeears/task/area. They were resampled to 0.0025° (~250 m) by nearest-neighbour using gdalwarp (GDAL/OGR contributors, 2020), and the burned area for temperate subregions was calculated from the union set of burned cells over the 12-month period in those areas. A small proportion of cells were flagged as burning more than once but were not recounted. The NASA-MODIS data are assumed to be capturing wildfires only and not low intensity prescribed burns.

The three annual burned area datasets used in this study are compared in Supplementary Fig. 1, and the longer two datasets used for trend analyses are also compared spatially in Supplementary Fig. 8.

**Years since the last fire**. We construct an annual 250 m gridded database of the number of years since the last wildfire (YSLF) occurrence at the end of the previous calendar year. It is based on the State and Territory fire histories from 1930 to 2019, considering all 0.0025° cells that burned at least once during the period and masking all others from the analysis. The fire histories data were completed to June 2020 by using burned cells for complete calendar years from monthly MODIS Burn Date data[72] aggregated annually. The number of years since the last fire (YSLF) is selected as the most robust variable to study trends over time because it covers the largest pixel-year population (pixels with at least one wildfire) and gives the current state of wildfire regime based on the last fire. This choice, with the data initialization method below, led to a total area of 155,384 km$^2$ used in the calculations from the total of 324,873 km$^2$ forest area considered in this study. We exclude prescribed burns to focus the analysis on the impacts and trends of changes in environmental conditions on fire activity (i.e., wildfires).

At 1930, the YSLF is unknown for all candidate cells except those burned during that year, which are set to zero. The rest of the 1930 YSLF grid is initialized with estimates by drawing randomly ($u$) from the geometric inverse distribution scaled by the mean wildfire frequency ($p$) (Eq. (1)) calculated from each pixel's fire history:

$$F^{-1}(u) = \left\lceil \frac{\ln(1-u)}{\ln(1-p)} \right\rceil 0 < u < 1 \tag{1}$$

For subsequent years, the estimated and known YSLF are incremented until historical fire events reset the YSLF to zero years. Therefore, over the entire period of the dataset, the YSLF surfaces go from mostly estimated in 1930 to entirely observed fire occurrence by 2019. This initialization method allows us to use a large population of pixel years with at least one fire since the start of the record in 1930, with observations of both fire occurrence and no-fire occurrence used in the calculations. We only provide the mean YSLF values for the last four decades when estimates have the most observations on fire occurrence and therefore are the most robust.

**Fire weather indices and climate/weather variables**. Forest Fire Danger Index. Fire weather data for near-surface conditions are examined here based on the FFDI[38], which is similar to a range of other fire weather indices used around the world and based on humidity, wind speed, temperature and rainfall[20,73,74]. The FFDI data used here are daily grids of 0.05 degrees in both latitude and longitude throughout Australia, available back to 1950[15]. The data are calculated from a gridded analysis of observations throughout Australia including for temperature, rainfall and humidity from the Australian Water Availability Project (AWAP) as well as wind speed from reanalysis data calibrated to the operational fire weather warning forecasts produced operationally by the Australian Bureau of Meteorology[15]. Rainfall data from AWAP are also presented in this study, as well as temperature data from the ACORN-SAT high-quality dataset for Australia.

Continuous-Haines Index. Fire weather data for higher levels above the surface are based on the C-Haines index which was adapted for Australian conditions[46] based on development of the Haines Index in North America[75]. The C-Haines index represents vertical atmospheric stability and humidity measures relating to pyroconvective processes, based on two subcomponents: a stability measure based on the temperature difference from 850 to 700 hPa and a humidity measure based on the 850 hPa dew point depression. It is calculated here using the method described in a previous study[40] which applied it to ERA-Interim reanalysis data from 1979 to 2016, with the data updated for this study from 1979 to the end of 2019 based on ERA5 reanalysis data.

Dry lightning. A data set containing dry lightning information for Australia was recently developed[53] based on a combination of rainfall based on observations and environmental conditions associated with thunderstorms based on reanalysis data. The data are gridded through Australia and available for each day during the time period from 1979 to 2016. The method used for thunderstorm environments is based on previous approaches developed in North America[76] and subsequently applied in Australia[77], but calibrated specifically for individual locations to reproduce the observed lightning frequency which can vary considerably throughout Australia[53].

Changes in these weather-related variables are presented from the first to the second half of the period used in this study, dependent on data availability for each individual variable. This is from 1980–1998 to 2000–2019 for temperature, rainfall, FFDI[15] C-Haines[40] and soil moisture (see Biomass and fuel loads), and from 1980–1999 to 2000–2016 for dry lightning[53].

**Biomass and fuel loads**. We performed continental simulations with the terrestrial biosphere model CABLE-BIOS[57,78] at 0.25° spatial resolution. The model was forced with Australian Bureau of Meteorology daily gridded rainfall, temperature, solar radiation, and vapour pressure, and near surface wind for the period 1900–2018. Except for rainfall, all observed time series started later than 1900. The gaps in the temperature (1900–1910), solar radiation (1900-1989), vapour pressure

(1900–1950) and wind (1900–1974) were filled with synthetic daily surfaces generated from 30-year monthly climatologies (29 years for solar radiation) from the nearest period to the missing data. Biomass (including wood and leaf components) and litter stocks (coarse woody debris, fine litter, and very fine litter) (all in kg C m$^{-2}$) were averaged spatially across all grid cells dominated by forest ecosystems and for each year (see "Methods": Forest extent and types). The aboveground fraction of woody biomass was calculated as a function of total shoot biomass according to[58], whereas all litter components were assumed to be 40% aboveground, which is in agreement with data presented in a meta-analysis[79]. For the analyses, we only use the aboveground fraction of the litter pool. Two different simulations allowed us to attribute the combined and individual contributions from the elevated $CO_2$ effect and changes in climate based on CABLE with TRENDY protocols (S1: $CO_2$ effect only), S2 ($CO_2$ + climate effect): S2-S1 (climate effect only)[59]. TRENDY (trends in the land carbon cycle) is a model inter-comparison of Dynamic Global Vegetation Models (DGVMs) with common model experiments and driving datasets. It is used to estimate the global land sink for the Global Carbon Project's annual update of the global carbon budget[59]. The protocol is fully described here https://sites.exeter.ac.uk/trendy/. CABLE-BIOS also simulated soil moisture to 23 cm in depth. CABLE provides potential biomass and fuel loads based on climate and $CO_2$ and their changes over time, but does not simulate previous fires or other disturbances that also affect fuel loads.

## Data availability

State and territory burned area data are made available in the online repository Zenodo[80] (https://doi.org/10.5281/zenodo.5631766). The area data are provided as one or more ArcGIS shapefiles per state and territory, and nationally for the National Indicative Aggregated Fire Extent Datasets (NIAFED). The original data sources are in Table 5 Supplementary Information.

The burned area based on NOAA's Advanced Very High-Resolution Radiometer (AVHRR)-Landgate burned area with continental coverage is available from https://myfirewatch.landgate.wa.gov.au. The NASA-MODIS burned area is accessible from https://lpdaacsvc.cr.usgs.gov/appeears/task/area. Climate, weather and biomass data is also available from the online repository[80] (https://doi.org/10.5281/zenodo.5631766). Contact the authors to access some of the high spatial resolution climate and weather data.

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

## Acknowledgements

Vale Vanessa Haverd, a co-author of this paper, colleague and friend, passed away during the time the manuscript was in review. We acknowledge and honour her great scientific legacy in understanding better how the global biosphere and Australian terrestrial eco-systems function and how they are being altered by environmental changes. This project was jointly funded through the Climate Science Centre of the Commonwealth Scientific and Industrial Research Organization (CSIRO), the Bureau of Meteorology (BOM), and the Earth Systems and Climate Change Hub of the Australian Government's National Environmental Science Program. We thank Claire Trenham for help with the MODIS data, and the internal reviewers of CSIRO and BOM Michael Grose, Richard Williams and Lynette Bettio.

## Author contributions

J.G.C., M.M., G.C., A.D. conceived and designed the work. M.M., G.C., A.D., P.R.B., J.K., A.P., V.H., J.G.C. analyzed data. All authors contributed to the writing and critical review of the paper.

## Competing interests

The authors declare no competing interests.
