## [Peer Review File · Nature Communications]

REVIEWER COMMENTS

Reviewer #1 (Remarks to the Author):

Overall comments

In this field there are many published articles on the topic of forest burned area and climate but they are often limited in scope, only focusing on a subset of the forest, or only analysing one or two fire metrics or limited by fire activity data. Whereas, this well written study brings together multiple variables that are critical to fire activity to explore the role of climate change on forested areas of Australia. This is achieved by combining a range of fire activity datasets (MODIS and NOAA-AVHRR derived, agency based and government datasets) to determine burned area and fire return interval trends. The study then analyses this in relation to changes in FFDI and separately explores changes in other relevant factors: C-Haines, dry lightning, temperature, precipitation and soil moisture. Finally, fuel loads are assessed in relation to prescribed burn records and using biophysical modelling.

I anticipated that these elements would be brought together by the end of the manuscript, however most of the variables were analysed and discussed in isolation. The trends in burned areas and return intervals will be of interest to many readers but they will be wanting to understand more about the spatial and temporal variability of these trends which is not presented.

The only variable that was analysed in relation to burned area was with FFDI and this identified a significant relationship with impressive predictive capability. It would have been useful for the reader to understand how the other variables related statistically to forest burned area and whether multivariate analysis would result in greater predictive capability.

I do not underestimate the effort of creating a long term burned area dataset and this will be of continuous use for research however I thought that only one final product would be used for the fire record but reading through the results it becomes confusing with all the different fire datasets and time periods being referred to. Furthermore, the differences between the datasets needs to be further assessed to assist the reader in determining the quality of each fire dataset.

The section on Fire Risk factors and links to fuel loads does not flow as well as the rest of the paper. Initially, the discussion leading into fuel loads makes the assumption of a possible increase in fuel load may be one of the factors that have resulted in upward trend in burned areas. The authors then propose two hypotheses to test (prescribed burning and climate change). This almost seems like a separate paper yet needs a lot more analyses and discussion around what the authors findings mean. The test for prescribed burns is simply looking at the trend of prescribed burns without considering many factors such as how reliable the data are and also the severity of those burns to remove fuel. The second test uses a biospheric model, CABLE, to model fuel load changes due to climate change (climate an CO2 changes) and finds increase in above ground biomass and coarse woody debris and a decline in fire litter associated with declining canopy leaf biomass. However, there is no analyses to link it back to the burned area and the discussion is not adequate. The fuel load is such an important aspect to fire activity but it has not been sufficiently linked to the rest of the paper.

In summary, this paper is well written and there is a real need in this field for a study that explores how climate change is linked to forest burned areas of Australia. This paper can be improved through further analyses between variables and an exploration of the influence of combinations of these variables as they are not contributing in isolation. Even a comprehensive discussion on the trends identified and how the variables interact would greatly improve the paper.

Specific comments

Abstract

Page 2, lines 3-4: The authors state that land management decision greatly affects fire activity in the first sentence of the abstract. I think that this would need to then be followed up with discussion in the main body of the paper.

Page 2, line 5: delete 'now' from the sentence

Main

Page 4, line 4: The paper cited here is misleading – this only covers the role of ENSO on FFDI not burned areas. There are papers linking ENSO and fire activity (including burned area) in south east Australia include:

Mariani, Fletcher, Holz and Nyman (2016) ENSO controls interannual fire activity in southeast Australia GRL 43(20)

Page 4, Line 15: The authors need to explain how/why they defined a mega fire as >100 000km²?

Page 4, Line 18-26: do the NOAA AVHRR trends align with the agencies and government fire datasets?

The readers would be interested to see what months and where these trends exist? This would give the reader a better appreciation of the spatial and temporal variability of the trends.

Page 4, lines 18-26: The paper focused on the March to August period having the largest trends in burned area - further analyses with the variables should be conducted to explain these trends.

Page 6, Lines 15-18: these statements about diminishing forests, land clearing and firefighting capability need referencing.

Page 7, lines 4-8: The finding of reduced fire return intervals will be of interest to many readers. Does severity play a role in the seed bank of Mountain Ash? If so it might be worth mentioning that the severity would be variable so we don't know if these locations are "hovering over the threshold of regional ecosystem collapse".

Page 8, Lines 5-6: I think change the 'is' to 'was' in this sentence about FFDI development. Noting it was developed in the 1960s using low intensity fires under mild conditions to determine suppression capability and has not been updated since.

Page 8, line 17-18: Useful finding that most of the years with severe or greater occurred since 2000. The readers would like to have those years listed somewhere. Also, worth referencing other papers that have compared FFDI and burned areas.

Page 8, Lines 20-25: Really useful - highlight of the paper.

Page 8, Lines 27 to page 9 lines 2: Interesting finding and would be good to ensure it is pointed out that some recent fires have burned large areas under lower FFDI values because of the pyro convective activity (driven by instability of atmosphere and fuel build up).

Page 8 Line 30 to page 9 line 2: The authors have given the predicted area burned but as I reader I need a reminder of what the actual area burned was for 2019/20 fire season so I can compare performance without needing to flick back over the paper.

Page 9, line 4: change 'clear' to 'clearly'

Page 11, lines 30 to page 12 line 1: Does the paper cited here actually contain information on lightning being responsible for 50% of area burned in some areas? I am not saying the value is

wrong just whether that is the most accurate source.

Page 13, Line 6: This is the first mention of an increase in fuel loads. Need greater discussion explaining the role of fuel load in fire activity before this section.

Page 13 lines 14-16: Should include cultural burning as another reason for conducting burns.

Page 13, Line 12-24: some readers will question the quality of the prescribed burning dataset analysed and the most important factor of a fuel reduction burn is whether it was successful or not, how much fuel was removed and how complete was that across the prescribed burn. I doubt these records capture the completeness of the prescribed burns.

Page 13, Line 19-22: I think the numbering of the figures is incorrect it should be figure 4 although I note there is no figure 5. Also note that these data only cover 1990-2015, not the four decades that are mentioned.

Page 14 line 16- page 15 line 1: add a reference to the statement that coarse fuels are more associated to fire severity.

Page 15, line 1-4: make it clear that soil moisture is only a proxy as I am not sure of the research that links soil moisture to overall forest system dryness.

Discussion and Conclusion

Page 16, Lines 4-5: a greater discussion required by the authors on what is driving the exponential growth of burned area in winter-spring period. Where and when are the greatest changes and why might this be the case when other studies indicate the greatest change in fire weather to be in spring and summer.

Page 16, line 8: change 'and' to 'an'

Page 16, lines 10-13: It would be useful for readers if analyses are conducted on multiple explanatory variables. The authors could predict burnt area using FFDI, can that be improved if dry lightning is included in the analyses?

Page 16, lines 18-19: I think claiming this study showed declining fire risk and increasing fire severity factors using biophysical modelling is overstepping the mark. The authors should rewrite to explain that the different fuel parameters the authors examined showed these trends and that these are one of the factors of fire risk and fire severity.

Methods

Page 19, Line 7-9: Note that the dataset is well validated for northern parts of Australian and commonly used but is less accepted in eastern and southern areas because of the coarse resolution.

Page 19, Lines 18-25: The authors mention the omission rate of prescribed burns but are there any studies that have analysed the omission rate for wildfires? It would be useful to include those result or if unavailable recognise the limitations of the dataset in capturing a proportion of the wildfires.

Page 20, Lines 11-13: The authors refer to omitting low-intensity prescribed fires but the authors should mention that this means low-intensity wildfires will be omitted.

Page 20, Lines 15-16: This reference to a large fire in Victoria is out of place. I don't understand why it has been placed here?

Pages 18-21: Three different fire datasets were used in this study –two derived from satellite products NOAA-AVHRR, MODIS and then another from fire and government agencies (which no

consistency between them). Was there any comparison between datasets? I assume there are large discrepancies and while this does not mean the results are not valid and useful at some scales it is important to recognise the issues.

Page 21, Lines 12-14: The authors state that their last fire occurrence database provides information on the fire frequency and fuel age but having omitted the planned burns means that fuel age is unlikely to be accurate.

Page 21-22, Lines 22-28: I do not know if this is a valid technique for determining fuel age without data. There were major fires in Victoria in 1926 that would play a large role in fuel age. However, the focus of analysis for this study is the last four decades so I assume this should resolve any issues.

Page 23, Line 14-16: I know very little about biomass and fuel load modelling but I do note the very coarse resolution of the output. How well validated is the output?

Reviewer #2 (Remarks to the Author):

The authors' investigated changes in annual burned area in forests of Australia using both a satellite-derived (AVHRR) burned area product (32 years) and integrated State and Territorial data (90 years). They conclude that the trend in burned area over time is increasing significantly and that the most likely cause is changing climate conditions. While the manuscript is very well written, per se, making it easy to read, erroneous inferences and crude average results for forests of the entire Australian continent detract from the strength of the findings and conclusions. That said, the authors have pulled together a very interesting dataset and could make this into a very informative manuscript of broad interest.

Specific points follow.

1. How are fire frequency, fire return interval, fire return period, fire return times, and mean fire return interval being defined/calculated? Some terms are being used synonymously and all assume the reader knows what is intended although different forest types and regional differences are being lumped together (apparently). Although spatial data and analyses (?) were done, decadal mean (?) FRIs are given for forests without any standard deviations. The methods (page 21 Lines 14-16) indicate only areas that burned at least one were used with those that were unburned masked out. If correctly described, this was incorrectly calculated. The fire return interval is for a given area of forest with the understanding that some areas burn more than once while others do not burn at all. As stated, it says that fires happened where forests burned and the rest of the forest don't matter. I am assuming that it was calculated correctly and just described erroneously. In any case, somewhere in the text, if not in Figure 2, there should be some quantification of just how much area of forest never burned during the period of analysis.

2. Provide some context of what the so-called natural fire regimes and fire return intervals are for the affected forests. Although the calculated fire return intervals are shortening by decade, the reported values are either still well above the fire regime characteristics for the majority of examined forests or comfortably within the range for even the tall eucalypt portion of the forests, as reported in the cited Murphy et al 2013 reference. Note, the listed forest types from the National Forest Inventory closely correlate with Table 1's B and C classes (Murphy et al. 2013). This is not to say that these fires are not creating massive social impacts but some ecological context is needed for the inferences being made.

3. Inaccurate inferences are being drawn from the data. Specifically, stating that "the rapidly decreasing fire return interval shown across forest ecosystems suggest the effect, if any, on fuel load would be a reduction given the shorter time intervals for forest to recover after fire" is

fallacious. The reasons being 1) despite the reduction in the mean FRI for forests, the fires themselves are spatial, with some areas more affected while others are less or unaffected (hence accumulating fuel), and 2) even the mean FRI values being reported for the more recent decades are greater than typical fire return intervals for most eucalypt forests and even within the interval for rare (~5%) extreme events (again Murphy et al. 2013). Therefore, the decreasing fire return intervals would reduce the average rate of fuel increase, not the mean fuel load across these forests unless something more can be said on increasing fuel consumption.

4. Similarly, the conclusion from rejecting the hypothesis (Ho1) about purported reduction in hazard reduction burns in forests (i.e. prescribed fires) due to the calculated lack of trend in area being burned is erroneous. As the authors state (Page 13 lines 18-19), the prescribed fires burn ~1% of forests annually. Even treating these fires as equivalent to wildfires in fuel consumption rates, this equates to roughly a ~100 year fire rotation interval which is much higher than what most of these forests typically experienced before fires were widely suppressed (e.g. <20 years for most). Therefore, on a landscape basis, fuel amounts could still be increasing, even if the amount of hazard reduction burning remains unchanged. This too would be spatially and temporally variant and the authors have the data in hand to provide this information.

5. The spatial maps of the AVHRR burning and State/Territory based burning need to be presented, ideally with some level of temporal (e.g. decadal) distribution as supplementary figures, if not in the main text. Furthermore, the hazard reduction burns versus wildfires should be shown for the State/Territory maps. The authors have done an excellent job in pulling this information together, they should show it to the readers and use it in the analyses.

6. What is/are the spatial scale(s) of the State/Territory datasets? All 250m? If I am reading this correctly, MODIS burned area (500m) was resampled to 250 m to match the composited States/Territory dataset to provide coverage of the most recent months/year but that the AVHRR data were not used for this analysis other than to help attribute the time (month) of burning. Given this granularity, at what scale(s) were the fire return intervals calculated? How was forest area resampled? Figure 2 shows a latitudinal progression northwards in decadal burning. Much more than a broad-brush calculation of average change for forests of the entire continent of Australia could and I suggest should be presented. This would better support the contention of climate change being the main forcing factor and also yield more actionable information for managers.

7. What happened to Western Australia? The text and figure 1 of the methods imply that it was part of this analysis but Figure 2 of the main text and the findings omit it. Was it used or not? I would suggest at least a table by State to relate the relevant information.

8. Page 4 lines 10-12. It is not clear why there are 2 area values here. Succinctly explain this so they don't have to wade through the methods to figure it out.

9. Page 4 Lines 18-23. This is perhaps the most novel finding as presented. It would be stronger yet if you could show how it has progressed latitudinally over the decades in line with climate change.

10. Page 6 Lines 15-18. Good point but what is the corresponding estimate in the amount of cleared forest? Better yet would be the estimate of forest clearing for the period of study.

11. Page 6 Lines 23-25. Tell the reader the scale of the "pixel" being referenced here.

12. Figure 2 needs an inset of Australia with an indication of what portion is being shown in these panels. Not everyone is from Oz. What happened to the forests of Tasmania? Figure 1 of the methods shows a lot more forest there than is apparent in this figure.

13. Fire return interval – Something akin to Figure 2 but depicting the spatial fire return intervals, as calculated, would be illustrative.

14. Page 13 Lines 8-10, Hypothesis 2. Fuel loads have increased over time (period). This is due to forest regrowth and vegetative production (NPP). In the absence of sufficient burning these increases will tend to happen regardless of an contribution from potential CO₂ fertilization. CO₂ fertilization effects may have increased rates of increase but nothing here provides context for how much that might be. Jiang et al (2020 Nature) have recently shown that these forests have some of the lowest responsiveness to CO₂ 'fertilization' of any forests globally, so some more context is needed and these hypotheses should be reframed/thought.

15. General comments. More needs to be done to make clear the assumptions that are being made in making these calculations. 1) all fires are the same (nothing about type (crown, surface) or intensity/severity), 2) all forests are the same (regardless of composition/type, canopy closure, height or age), 3) mortality and fuel consumption are always 100%, resetting fuel age/loading to zero (and all fuels are the same regardless of size, location (surface, canopy, ground), live or dead, or moisture content (e.g. availability)), 4) climate change and weather are even across the continent. Simplifying assumptions are always needed but should be clearly stated, justified to the extent possible, and not be overly simplistic.

16. Page 23 lines 25-28. It is unclear what to make of this. Litter components should be on the surface of the ground or is this class mixing live foliage and leaf litter? 40% aboveground? Is the other 60% decomposed into humus? The citation given was for fine roots but may have quantified other elements. More clarity needed.

17. Figure 2-SI – needs a corresponding figure of the prescribed (hazardous reduction) burn information.

Reviewer #3 (Remarks to the Author):

Overall, this is an interesting, novel and timely analysis relevant to the journal Nature Communications. The analysis is multidimensional in that it explores temporal patterns in increases in forest area burned, along with relevant explanatory mechanisms.

While the analyses of temporal patterns in area burned are sound, the analyses of temporal patterns in fire return interval and some of the explanatory variables are less convincing. For example, the methods for fire return interval do not clearly differentiate between time since fire and interval between fires, with the latter metric being easily the most appropriate dependent variable for analyses. This is compounded by using synthetic data for fuel age whereas both the figures and analyses should be based only on observed intervals between fires.

There are also considerable synthetic elements, in some of the explanatory meteorological and associated variables, which requires further explanation in order to understand the effect of this approach on the study findings.

Finally, the hypothesis relating to CO₂ effects on fuel loads is secondary to a more appropriate hypothesis about what is the combined effect of climate and CO₂ effects on fuel loads over time.

Further specific comments on aspects of the manuscript:

Page 4, lines 13-16: It is not sufficiently justified why so-called 'megafire' years are variously included or excluded in analyses and associated commentary. If subsequent commentary explores the effect of not including 2002 and 2006, then why are they included in the analysis?

Page 6, lines 15-18: A reference for diminishing extent of forest over the study period, not the period since European invasion, would enhance rigor.

Page 7, line 13: The figure caption and associated text refers to times since fire, whereas the section heading is fire return interval. The inclusion in the text, figure and methods (page 21, lines 12 - 13) of time since fire is confusing and somewhat meaningless in this context and the narrative and figures should be based on actual intervals between fires.

Page 8, line 6: McArthur's FFDI indicates weather conditions across the 'Danger' spectrum, not just dangerous weather conditions.

Page 8, line 8: McArthur's FFDI incorporates a drought index and a 'Drought Factor', not a fuel moisture deficit.

Page 13, lines 9-10: There is another hypothesis regarding climate change effects on fuel loads, being that fuel loads are declining due to decreased vegetative productivity associated with drier conditions.

'Figure 4' appears twice and Figure 5 caption is missing.

Page 14, line 2: 'prescribed burnings' should be written as 'prescribed burning'. Same for page 19, line 19.

Figure 4, the second one: There does not appear to be any orange-coded data indicated. If it is too limited in area to be visible in the figure then that should be indicated in the figure caption.

Page 15, lines 6-10: Overall, if fine litter is projected to decline (Fig 6b), it is largely irrelevant for this study if the CO₂ effect in isolation is to increase aboveground biomass.

Page 16, line 15 to page 17, line 2: A paragraph of the conclusion is effectively repeated.

Some hyperlinks in the document are broken, e.g. Page 20, lines 30-31.

Page 21, lines 2-3: The sentence does not make sense.

Page 21, lines 22-30: The figures and analyses should be limited to realized fire intervals, without including any aspect of synthetic fuel age. Given the clarification on page 22, line 2 then the method should just state the analysis is based on observations, without invoking the synthetic estimates of fuel age in the manuscript.

Page 23, lines 19-22: Again, the reliance on synthetic data is not convincing for this application.

Table 1 - S1: The table needs to be carefully checked. There are several repeated pairs of dependent and independent variables (e.g. numbers 7 and 9; numbers 8 and 10).

REVIEW RESPONSES

REVIEWER COMMENTS

Reviewer #1 (Remarks to the Author):

Overall comments

In this field there are many published articles on the topic of forest burned area and climate but they are often limited in scope, only focusing on a subset of the forest, or only analysing one or two fire metrics or limited by fire activity data. Whereas, this well written study brings together multiple variables that are critical to fire activity to explore the role of climate change on forested areas of Australia. This is achieved by combining a range of fire activity datasets (MODIS and NOAA-AVHRR derived, agency based and government datasets) to determine burned area and fire return interval trends. The study then analyses this in relation to changes in FFDI and separately explores changes in other relevant factors: C-Haines, dry lightning, temperature, precipitation and soil moisture. Finally, fuel loads are assessed in relation to prescribed burn records and using biophysical modelling.

I anticipated that these elements would be brought together by the end of the manuscript, however most of the variables were analysed and discussed in isolation. The trends in burned areas and return intervals will be of interest to many readers but they will be wanting to understand more about the spatial and temporal variability of these trends which is not presented.

The only variable that was analysed in relation to burned area was with FFDI and this identified a significant relationship with impressive predictive capability. It would have been useful for the reader to understand how the other variables related statistically to forest burned area and whether multivariate analysis would result in greater predictive capability.

We have now performed multivariate analyses using the climate variables studied of FFDI, C-Haines and dry lightning. The main text reflects this extension from one variable analysis, and shows that both FFDI25 and FFD50 explain more than 2/3 of the burned area and that CHaines and dry lightning only improves marginally the explanatory power (probably because they are highly correlated, even if they are not mechanistically). See also supplementary Table 3 and supplementary figure 7. Potential simulated biomass fuels are not used in the multivariate analysis as the model does not reset loads after disturbances and show no correlation because are potential quantities (supplementary Figure 7).

I do not underestimate the effort of creating a long term burned area dataset and this will be of continuous use for research however I thought that only one final product would be used for the fire record but reading through the results it becomes confusing with all the different fire datasets and time periods being referred to. Furthermore, the differences between the datasets needs to be further assessed to assist the reader in determining the quality of each fire dataset.

We have made an effort to make very clear which data are being used and the justification of why is being used in each of the sections and Methods. Because the characteristics of each of the data products, we can extract different information from each one providing

complementary information. The AVHRR data product covers 32 years and it is methodological consistent across time and across all states and territories covered in this study. This makes the AVHRR product very appropriate to study trends over the past three decades because of its robustness and methodological consistency. The data from State and Territory agencies extending 90 years back in time enables to construct a longer-term view of the fire history of Australian forests, albeit there is no methodological consistency across the states and over time, and we don't know how accurate they are, particularly for the first half of the record. Cognisant of these limitations, we primarily use the States and Territory dataset to build a history of the fire return interval which can only be developed with a very long time series. However, we only provide results for the past four decades (to overlap with the AVHRR record) which the data is likely to be most complete. We use MODIS burned area to fill gaps and have an additional estimate for the most recent period. To show the similarities and deviations of the three products, we now show a key figure where we compare the three products, which shows remarkable consistencies for the major characteristics of the multidecadal trend, albeit with differences too. See Supplementary figures 1 and 8. In addition, we have added a number of more detail figures using both main datasets for the reader to have more spatial and temporal resolution, eg, see figure 2, and supplementary figures 2, 3, 4.

The section on Fire Risk factors and links to fuel loads does not flow as well as the rest of the paper. Initially, the discussion leading into fuel loads makes the assumption of a possible increase in fuel load may be one of the factors that have resulted in upward trend in burned areas. The authors then propose two hypotheses to test (prescribed burning and climate change). This almost seems like a separate paper yet needs a lot more analyses and discussion around what the authors findings mean. The test for prescribed burns is simply looking at the trend of prescribed burns without considering many factors such as how reliable the data are and also the severity of those burns to remove fuel. The second test uses a biospheric model, CABLE, to model fuel load changes due to climate change (climate and CO₂ changes) and finds increase in above ground biomass and coarse woody debris and a decline in fire litter associated with declining canopy leaf biomass. However, there is no analyses to link it back to the burned area and the discussion is not adequate. The fuel load is such an important aspect to fire activity but it has not been sufficiently linked to the rest of the paper.

We have extended the discussion for the fuel load section, and importantly, we have clearly spell out the limitations of the data and what conclusions can be drawn from the two analyses. We agree with the reviewer that the prescribed burning analysis has no direct information on the changes in fuel loads as data on fuel load reduction is not available. We have also nuanced the fact that our modelled fuel loads and biomass are potential quantities and changes because the model doesn't keep track and resets loads for each previous fire and because the coarse resolution of the model. We think the information is valuable mainly to response to the potential role of climate change, but we provide no direct evidence on the role of changes in fuel loads in fire risk.

In summary, this paper is well written and there is a real need in this field for a study that explores how climate change is linked to forest burned areas of Australia. This paper can be improved through further analyses between variables and an exploration of the influence of combinations of these variables as they are not contributing in isolation. Even a comprehensive discussion on the trends identified and how the variables interact would greatly improve the paper.

We hope that the additional analyses and discussion, along with a much more detail set of figures will address the reviewer's comment.

Specific comments

Abstract

Page 2, lines3-4: The authors state that land management decision greatly affects fire activity in the first sentence of the abstract. I think that this would need to then be followed up with discussion in the main body of the paper.

We have removed the sentence as we realize we opened an important and large topic on effects on fire activity that is outside of the scope and data of this analysis.

Page 2, line 5: delete 'now' from the sentence

Done

Main

Page 4, line 4: The paper cited here is misleading – this only covers the role of ENSO on FFDI not burned areas. There are papers linking ENSO and fire activity (including burned area) in south east Australia include:

Mariani, Fletcher, Holz and Nyman (2016) ENSO controls interannual fire activity in southeast Australia GRL 43(20)

The revised text and provide more detail, including greater clarification of the influence of ENSO on fire weather which can then influence fire occurrence and burned area. We included the reference suggested, and also Mariani and Fletcher 2016, GRL, which covers a second important climate mode: The Southern Annular Mode.

Page 4, Line 15: The authors need to explain how/why they defined a mega fire as >100 000km²?

We use the term "megafire year" to indicate the top 10 percentile years with most burned area in the 32-year AVHRR record. The 100,000 km² is a rounding of the minimum area burned on those top 10 percentile years. We have now made clear how it is defined in our study.

We don't use the term "mega fire" because of its complexity to define it as it requires a number of assumptions on what constitute a single mega fire as opposed to the evolution of a fire complex.

Page 4, Line 18-26: do the NOAA AVHRR trends align with the agencies and government fire datasets?

The readers would be interested to see what months and where these trends exist? This would give the reader a better appreciation of the spatial and temporal variability of the trends.

We show Supplementary figure 1 with a comparison of the three datasets used in this study: AVHRR, State and Territory agencies, and MODIS. It shows a remarkable consistent of the main features of the multi-decadal trends, but as expected, also with important differences between them. In addition, we have created a number of new figures to provide insights into temporal and spatial dynamics of the main two datasets: AVHRR and Agencies data: figure 2, and supplementary figures 2, 3, and 4.

Page 4, lines 18-26: The paper focused on the March to August period having the largest trends in burned area - further analyses with the variables should be conducted to explain these trends.

We now provide more discussion and figures to unrevealed what is contributing to the exponential growth of burned area in that period. We have decomposed the period in the two seasons, in two-months periods, and monthly. We show that Autumn contributed most in relative terms followed by Winter, but also make clear that in terms of absolute contributions of burned area, spring and summer were the most important for the increasing trend. The new figures are Figure 2, and Supplementary figures 2 and 3.

Page 6, Lines 15-18: these statements about diminishing forests, land clearing and firefighting capability need referencing.

Added:

- Marco Calderon-Loor, Michalis Hadjikakou, Brett A. Bryan (2020) High-resolution wall-to-wall land-cover mapping and land change assessment for Australia from 1985 to 2015. *Remote Sensing of Environment* 252 (2021) 112148
- NSW Government. NSW Rural Fire Services – History. <https://www.rfs.nsw.gov.au/about-us/history>. Accessed 22 December 2020

Page 7, lines 4-8: The finding of reduced fire return intervals will be of interest to many readers. Does severity play a role in the seed bank of Mountain Ash? If so, it might be worth mentioning that the severity would be variable so we don't know if these locations are "hovering over the threshold of regional ecosystem collapse".

We agree that a fire return interval below 20 years in mountain ash forests, unless with high severity fires leading to high tree mortality, it is not necessarily a condition leading to ecosystem collapse. Unfortunately, we have no concurrent historical information on fire severity. We have reworded the text about the significance of shortening fire return intervals (mean years since the last fire in our study) and qualify the use of ecosystem collapse. The text has been moved to the Discussions and Conclusion section. "Together these trends have led to a rapid shortening of the fire return interval in forest ecosystems across Australia as represented in our study by years since last fire (YSLF). The mean YSLF values across the last four decades are still within the historical fire regimes reported for the same type of forests⁹, however, pixel level analysis reveals that fire return interval in some areas is already outside of historical fire regimes (Figure 4). Particularly, mountain ash (*Eucalyptus regnans*) and alpine ash (*Eucalyptus delegatensis*) forests in Victoria are among the most vulnerable ecosystems. Both species are obligate seeders that require 20-30 years to reach maturity and begin to establish a seedbank able to recover the forest after fire⁵⁴⁻⁵⁶. Some regions in Figure 4 shows mean YSLF of about 20 years or less for the last decade. At this short fire return intervals, if fires were to be of high severity leading to tree mortality (which this study has not addressed), both ecosystems types would be at risk of collapse."

Page 8, Lines 5-6: I think change the 'is' to 'was' in this sentence about FFDI development. Noting it was developed in the 1960s using low intensity fires under mild conditions to determine suppression capability and has not been updated since.

Corrected.

Page 8, line 17-18: Useful finding that most of the years with severe or greater occurred since 2000. The readers would like to have those years listed somewhere. Also, worth referencing other papers that have compared FFDI and burned areas.

We have added the specific years in the page and line, and reference Bradstock et al. 2009.

Page 8, Lines 20-25: Really useful - highlight of the paper.
Thanks

Page 8, Lines 27 to page 9 lines 2: Interesting finding and would be good to ensure it is pointed out that some recent fires have burned large areas under lower FFDI values because of the pyro convective activity (driven by instability of atmosphere and fuel build up).
We have added additional text and the most notable example of the Canberra fires in 2003. We have also done a new multivariate analysis that shows that the C-Haines does not improve in a significant way the explanatory power of FFDI, mostly because they are also highly correlated, despite being driven by different processes.

Page 8 Line 30 to page 9 line 2: The authors have given the predicted area burned but as I reader I need a reminder of what the actual area burned was for 2019/20 fire season so I can compare performance without needing to flick back over the paper.
Added.

Page 9, line 4: change 'clear' to 'clearly'
Changed

Page 11, lines 30 to page 12 line 1: Does the paper cited here actually contain information on lightning being responsible for 50% of area burned in some areas? I am not saying the value is wrong just whether that is the most accurate source.
The reference (#35 in first submitted ms.) was changed to Dowdy, A.J. and Mills, G.A., 2012. Characteristics of lightning-attributed wildland fires in south-east Australia. *International Journal of Wildland Fire*, 21(5), 521-524. This new reference provides the details on the area burned > 50% in this region by lightning-ignited fires. The relevant text from that study is: "Lightning-fires account for ~90% of the total area burnt during the available period of data, even though they only account for ~30% of the total number of fires that occurred. This disproportionality can partly be attributed to the Victorian Alpine fires of January–March 2003, which burnt $\sim 1 \times 10^6$ ha, although even without this one large event, lightning-fires still account for a high proportion (~55%) of the total area burnt by all fires."

Page 13, Line 6: This is the first mention of an increase in fuel loads. Need greater discussion explaining the role of fuel load in fire activity before this section.
The section "Fire risks factors and fuel loads" now begins with an explanation of the role and importance of fuel loads in fire activity:
"In addition to weather and climatic conditions, and the availability of ignition sources for a fire to occur, continuous biomass sufficiently dry to burn is also required for a fire to take place and spread^{46,47}. The availability of fuel plays a key role in determining the intensity (peak energy output) and severity (impacts as related to the amount of canopy scorch, canopy loss, tree death, biomass loss) of fires. Understanding the role of fuel loads in fire activity is important as it also provides the knowledge base to reduce fire risk through the management of fuel quantity and distribution in the landscape^{46,48}. Fuel loads composed of aboveground biomass (particularly leaf biomass) and litter, including coarse woody debris (e.g., dead branches, logs, standing dead trees) and fine litter (eg, leaf, twigs)."

Page 13 lines 14-16: Should include cultural burning as another reason for conducting burns.
Added: "... , and cultural burns by indigenous people".

Page 13, Line 12-24: some readers will question the quality of the prescribed burning dataset analysed and the most important factor of a fuel reduction burn is whether it was successful or not, how much fuel was removed and how complete was that across the prescribed burn. I doubt these records capture the completeness of the prescribed burns.

We agree with the reviewer on the importance to know the actual amount of fuel reduction achieved with the prescribed burns. Unfortunately, we don't have such a data and our understanding is that it is not routinely acquired, and certainly not for the the study period and continental forest coverage of this study. We have now added a sentence in Methods "State and Territory fire histories data" where we acknowledge the lack of information on the actual fuel reduction achieved, and we have also reviewed our conclusions in light of this uncertainty (in "Discussion and Conclusions")

Page 13, Line 19-22: I think the numbering of the figures is incorrect it should be figure 4 although I note there is no figure 5. Also note that these data only cover 1990-2015, not the four decades that are mentioned.

Thanks, it was a misnumbering of the figures which we have now corrected. We have also provided the exact time period that the figure covers, 1988 to 2018, to match and be comparable with the 32-year record of the AVHRR burned area dataset. 2019 data on prescribed burning area is not available yet.

Page 14 line 16- page 15 line 1: add a reference to the statement that coarse fuels are more associated to fire severity.

We have extended the sentence and provide a reference: "Fine fuels are key determinants of fire risk and initial spread⁵⁴, while coarse fuels are more associated with total energy output and overall fire severity⁴⁴."

Page 15, line 1-4: make it clear that soil moisture is only a proxy as I am not sure of the research that links soil moisture to overall forest system dryness.

We reworded the paragraph and we no longer make a direct link between soil dryness trends and overall forest dryness. We also include the word "proxy": "Decrease soil moisture down to 23 cm depth over the past 40 years, as simulated by CABLE and as a proxy for reduced water availability to the forest floor (Fig. 4f) (see Methods: Biomass and fuel loads), suggests a likely increase in dryness of fine litter, which increases fire risk⁵⁴."

Discussion and Conclusion

Page 16, Lines 4-5: a greater discussion required by the authors on what is driving the exponential growth of burned area in winter-spring period. Where and when are the greatest changes and why might this be the case when other studies indicate the greatest change in fire weather to be in spring and summer.

We have new discussion on the characteristics of the exponential growth, both in Discussion and also in the Results section. We show that while the higher relative growth has been during the autumn and winter, the absolute contributions to the increase in burn area are in fact in spring and summer, consistent with previous work and reviewer comment:

"These results indicate an extension of the fire season into the cooler months of the year, with over a five-fold increase in the annual mean burned area in winter and three-fold increase in autumn between the first and second half of the studied period. However, both spring and summer contributed about 10 times more to the increase in burned area than autumn and

winter (Fig. 2; Supplementary Fig. 2 and 3). The seasonal fractions of burned area for the 32-year period are 62% in summer, 28% in spring, 7% in autumn and 3% in winter.”

We have also included new discussion on latitudinal temperature gradient, south to north, which provides further insights into where the contributions to the increase trend are coming.

Page 16, line 8: change ‘and’ to ‘an’
Done

Page 16, lines 10-13: It would be useful for readers if analyses are conducted on multiple explanatory variables. The authors could predict burnt area using FFDI, can that be improved if dry lightning is included in the analyses?

We have done a new multivariate regression analyses with all fire weather variables including FDDI25, FDDI50, C-Haines and dry lightning. See key paragraph below and Supplementary Table 3 for the regression parameters and supplementary figure 8.

Page 16, lines 18-19: I think claiming this study showed declining fire risk and increasing fire severity factors using biophysical modelling is overstepping the mark. The authors should rewrite to explain that the different fuel parameters the authors examined showed these trends and that these are one of the factors of fire risk and fire severity.

We have removed this conclusion and qualify that our potential fuel related analysis does not provide the evidence required to say anything about changes in fire intensity or severity. Our modelled potential quantities are affected by climate, but not by the land use history and fire of each single cell; the estimates are also of low resolution and not matching well the rest of high-resolution data. We think the analysis is still informative to explore the potential role of climate change. Our revised conclusions reflect this view.

Methods

Page 19, Line 7-9: Note that the dataset is well validated for northern parts of Australian and commonly used but is less accepted in eastern and southern areas because of the coarse resolution.

Noted and highlighted the key reference that validated the dataset for forests in the south-eastern region.

Page 19, Lines 18-25: The authors mention the omission rate of prescribed burns but are there any studies that have analysed the omission rate for wildfires? It would be useful to include those result or if unavailable recognise the limitations of the dataset in capturing a proportion of the wildfires.

We agree with the review that the AVHRR product is likely to omit all low intensity fires, not only prescribed burns, but also low intensity wildfires. We are not aware of any study that has looked into this issue, and we now acknowledge this limitation in the text in Methods where we described the dataset.

Page 20, Lines 11-13: The authors refer to omitting low-intensity prescribed fires but the authors should mention that this means low-intensity wildfires will be omitted.

We now mention that the AVHRR data is likely to miss low intensity wildfires too.

Page 20, Lines 15-16: This reference to a large fire in Victoria is out of place. I don’t understand why it has been placed here?

We have removed it.

Pages 18-21: Three different fire datasets were used in this study –two derived from satellite products NOAA-AVHRR, MODIS and then another from fire and government agencies (which no consistency between them). Was there any comparison between datasets? I assume there are large discrepancies and while this does not mean the results are not valid and useful at some scales it is important to recognise the issues.

This answer is a repeat of the answer provided at the beginning of the review under the general comments:

“We have made an effort to make very clear which data are being used and the justification of why is being used in each of the sections and Methods. Because the characteristics of each of the data products, we can extract different information from each one providing complementary information. The AVHRR data product covers 32 years and it is methodological consistent across time and across all states and territories covered in this study. This makes the AVHRR product very appropriate to study trends over the past three decades because of its robustness and methodological consistency. The data from State and Territory agencies extending 90 years back in time enables to construct a longer-term view of the fire history of Australian forests, albeit there is no methodological consistency across the states and over time, and we don’t know how accurate they are, particularly for the first half of the record. Cognisant of these limitations, we primarily use the States and Territory dataset to build a history of the fire return interval which can only be develop with a very long time series. However, we only provide results for the past four decades (to overlap with the AVHRR record) which the data is likely to be most complete. We use MODIS burned area to fill gaps and have an additional estimate for the most recent period. To show the similarities and deviations of the three products, we now show a key figure where we compare the three products, which shows remarkable consistencies for the major characteristics of the multidecadal trend, albeit with differences too. See Supplementary figures 1 and 8. In addition, we have added a number of more detail figures using both main datasets for the reader to have more spatial and temporal resolution, eg, see figure 2, and supplementary figures 2, 3, 4.”

Page 21, Lines 12-14: The authors state that their last fire occurrence database provides information on the fire frequency and fuel age but having omitted the planned burns means that fuel age is unlikely to be accurate.

We realize we used the various terms of fire frequency, fuel age and fire return interval a bit lose leading to confusion. We have now explained more precisely and extensively what we did and calculated with the deletion of the differet terms which indicate different things. Our analysis estimates the number of years since the last fire (YSLF) for all cells with at least one wildfire in the State and Territory agency record. This choice is justified because the focus of the ms. is to study multidecadal trends over time. The focus on YSLF for cells with fire occurrence data enabled to do this study most quantitatively and robustly, and cover the largest pixel-year population. We justify the choices further in Methods. A fire return interval of the cells burned at least once is likely to be correlated with the broader fire frequency of a region or vegetation class, but it is not the same. We think the revised explanation in the main text and methods is clearer, helped by the removal of multiple terms used before.

Page 21-22, Lines 22-28: I do not know if this is a valid technique for determining fuel age without data. There were major fires in Victoria in 1926 that would play a large role in fuel age. However, the focus of analysis for this study is the last four decades so I assume this should resolve any issues.

The purpose of the method used is to be able to take advantage of the full population of observations since the very beginning of the record in 1930. For instances, if a particular pixel has its first fire eg, in 1980 and a second one in 2006, by only using the second fire to determine a fire return interval, we would dismiss 50 years of observations (1930-1979) in that pixel that a fire had not occurred. Because we know that fires had occur sometime in the past before 1930 (as per the fire history of Australia and paleo records), our method applies a probability distribution function based on the pixels with fire occurrence over the 90-year period to determine when a fire might have occurred prior to 1930. Thus, by the time we arrive to 1930, each pixel/year is being used as an observation (fire or no fire). Once the first fire occurs in a pixel, then the calculations proceed based on the changes in YSLF of that very pixel. By the time we reach the last four decades for which we provide results in the ms., the calculations are based on the fires that occurred in the pixel.

Page 23, Line 14-16: I know very little about biomass and fuel load modelling but I do note the very coarse resolution of the output. How well validated is the output?

CABLE fuel loads have been benchmarked with observations with a dedicated paper, now clearly cited in the text: Volkova et al 2018: A data - Model fusion methodology for mapping bushfire fuels for smoke emissions forecasting in forested landscapes of south-eastern Australia.

Having said that, we agree with the reviewer that the coarse resolution of the model output limits the value of the fuel load findings, particularly when used with data at a much higher resolution for the analyses. We have rewritten the conclusions from the biomass analysis to be more nuance and make clearer the data limitations. We think that the simulations presented are best used to explore the possible impacts of climate change, but do not allow us to make a direct connection to changes in the actual fire risk.

Reviewer #2 (Remarks to the Author):

The authors' investigated changes in annual burned area in forests of Australia using both a satellite-derived (AVHRR) burned area product (32 years) and integrated State and Territorial data (90 years). They conclude that the trend in burned area over time is increasing significantly and that the most likely cause is changing climate conditions. While the manuscript is very well written, per se, making it easy to read, erroneous inferences and crude average results for forests of the entire Australian continent detract from the strength of the findings and conclusions. That said, the authors have pulled together a very interesting dataset and could make this into a very informative manuscript of broad interest.

Specific points follow.

1. How are fire frequency, fire return interval, fire return period, fire return times, and mean fire return interval being defined/calculated? Some terms are being used synonymously and all assume the reader knows what is intended although different forest types and regional differences are being lumped together (apparently). Although spatial data and analyses (?) were done, decadal mean (?) FRIs are given for forests without any standard deviations. The methods (page 21 Lines 14-16) indicate only areas that burned at least one were used with those that were unburned masked out. If correctly described, this was incorrectly calculated. The fire return interval is for a given area of forest with the understanding that some areas burn more than once while others do not burn at all. As stated, it says that fires happened where forests burned and the rest of the forest don't matter. I am assuming that it was calculated correctly and just described erroneously. In any case, somewhere

in the text, if not in Figure 2, there should be some quantification of just how much area of forest never burned during the period of analysis.

We agree with the reviewer that we had used interchangeably some of the terminology which are not equivalent. That led to some confusion as to what exactly the analysis was doing, the intent of the analysis and therefore, its conclusions.. We have now rewritten what we have done and justify better our choices and the exclusive use of “years since the last fire, YSLF”. Our analysis was done for each of the pixels that burned with a wildfire at least once, and omitted the cells that never burned in the 90-year database. We did not include prescribed burns because the focus of the analysis is to show how environmental factors have changed trends in wildfire burned area, excluding the direct human intervention thru fuel management in prescribed burning. This focus and choice also make clear we do not provide a fuel age. This approach allowed us to provide a quantitative analysis using the actual data available and without having to make any assumptions on those cells that had never burned (ie, how much longer than 90 years the fire return interval was, or did they ever burned?). Therefore, we are carefully to say that we are not estimating fire frequencies or fuel ages for various regions or vegetation types, but the number of years since last fire for the cells we have actually data on it.

Please see both the section in the main text and also Methods. We have also added the total forest study area and the burned fraction (Methods: Years since last fire).

2. Provide some context of what the so-called natural fire regimes and fire return intervals are for the affected forests. Although the calculated fire return intervals are shortening by decade, the reported values are either still well above the fire regime characteristics for the majority of examined forests or comfortably within the range for even the tall eucalypt portion of the forests, as reported in the cited Murphy et al 2013 reference. Note, the listed forest types from the National Forest Inventory closely correlate with Table 1’s B and C classes (Murphy et al. 2013). This is not to say that these fires are not creating massive social impacts but some ecological context is needed for the inferences being made.

Thanks for the suggestion to provide context to the results of the analysis. We have added the following text:

“For context, some of dominant vegetation types such as tall eucalypt forests have typical fire intervals of 20 to 100 years with extreme cases with more than 100 years⁹. Lower stature eucalypt forests have typical fire intervals between 5 and 20 years, with extreme cases with a range between 20 and 100 years⁹. Tropical forest all have fire return intervals above 100 years⁹.”

3. Inaccurate inferences are being drawn from the data. Specifically, stating that “the rapidly decreasing fire return interval shown across forest ecosystems suggest the effect, if any, on fuel load would be a reduction given the shorter time intervals for forest to recover after fire” is fallacious. The reasons being 1) despite the reduction in the mean FRI for forests, the fires themselves are spatial, with some areas more affected while others are less or unaffected (hence accumulating fuel), and 2) even the mean FRI values being reported for the more recent decades are greater than typical fire return intervals for most eucalypt forests and even within the interval for rare (~5%) extreme events (again Murphy et al. 2013). Therefore, the decreasing fire return intervals would reduce the average rate of fuel increase, not the mean fuel load across these forests unless something more can be said on increasing fuel consumption.

We agree with the reviewer and we have removed the pertaining statements from the text, and carefully qualify the conclusions inferred from the fuel analysis.

4. Similarly, the conclusion from rejecting the hypothesis (Ho1) about purported reduction in hazard reduction burns in forests (i.e. prescribed fires) due to the calculated lack of trend in area being burned is erroneous. As the authors state (Page 13 lines 18-19), the prescribed fires burn ~1% of forests annually. Even treating these fires as equivalent to wildfires in fuel consumption rates, this equates to roughly a ~100-year fire rotation interval which is much higher than what most of these forests typically experienced before fires were widely suppressed (e.g. <20 years for most). Therefore, on a landscape basis, fuel amounts could still be increasing, even if the amount of hazard reduction burning remains unchanged. This too would be spatially and temporally variant and the authors have the data in hand to provide this information.

As in point 3, we agree with the reviewer and we have removed the pertaining statements from the text, and carefully qualify the conclusions inferred from the fuel analysis

5. The spatial maps of the AVHRR burning and State/Territory based burning need to be presented, ideally with some level of temporal (e.g. decadal) distribution as supplementary figures, if not in the main text. Furthermore, the hazard reduction burns versus wildfires should be shown for the State/Territory maps. The authors have done an excellent job in pulling this information together, they should show it to the readers and use it in the analyses. We have included a number of new figures showing both spatial and temporal changes of burned area using the AVHRR and State/Territory datasets as the two main datasets, and include a figure where we compared all three datasets used in this study:

- Supplementary Figure 1. Comparison of wildfire burned area (km²) based on state and territory agencies, NOAA-AVHRR Landgate, and NASA-MODIS for the period 1988-2019 fire years.
- Supplementary Figure 2. Seasonal trends of burned area in km². Data: AVHRR-Landgate (1988-2018) and MODIS (2019).
- Supplementary Figure 3. Bi-monthly trends of burned area in km². Data: AVHRR-Landgate (1988-2018) and MODIS (2019).
- Supplementary Figure 4. Burned area (km²) of wildfires and prescribed burns for 1960 to 2019 fire year. Data: State and Territory agencies (1960-2018) and MODIS (2019).
- Supplementary Figure 8. Spatial distribution of burned area (km²) using NOAA-AVHRR Landgate and State/Territory agency data for 1988-2018.

6. What is/are the spatial scale(s) of the State/Territory datasets? All 250m? If I am reading this correctly, MODIS burned area (500m) was resampled to 250 m to match the composited States/Territory dataset to provide coverage of the most recent months/year but that the AVHRR data were not used for this analysis other than to help attribute the time (month) of burning. Given this granularity, at what scale(s) were the fire return intervals calculated? How was forest area resampled? Figure 2 shows a latitudinal progression northward in decadal burning. Much more than a broad-brush calculation of average change for forests of the entire continent of Australia could and I suggest should be presented. This would better support the contention of climate change being the main forcing factor and also yield more actionable information for managers.

We have provided additional explanations on how the various datasets were used.

The following has been added to Methods: State and Territory fire histories data.

“The agency data is provided as polygons in vector files derived from a variety of data that are converted to rasters at 0.0025deg resolution. Data includes paper records, ground mapping, and aerial photography for the earlier part of the record, and GPS boundary tracing

from ground surveys, line scans, aerial photography and in some cases satellite mapping for the later part of the record”

Yes, we resampled MODIS 500m to 250 to match the data above, and the years since last fire (YSLF) were calculated at 250m scale. We have made this clear both in the main text and in Methods.

As per latitudinal information, we have done additional analyses and provide text in the section, and include the new Supplementary Table 2, with the YSLF by state enabling to see the changes for the last four decades along a temperature gradient: Tasmania, Victoria, NSW+ACT and Queensland, in addition to WA and SA.

7. What happened to Western Australia? The text and figure 1 of the methods imply that it was part of this analysis but Figure 2 of the main text and the findings omit it. Was it used or not? I would suggest at least a table by State to relate the relevant information.

We have added a new continental-wide figure where we show all forest regions, one panel per each of the last four decades in Supplementary Figure 5.

8. Page 4 lines 10-12. It is not clear why there are 2 area values here. Succinctly explain this so they don't have to wade through the methods to figure it out.

The two numbers are the estimates based on 1) the state and territory agencies data, and 2) the NASA-MODIS burned area product. These are the two available datasets for the 2019 fire year. We have now made explicit what each the of the number is.

9. Page 4 Lines 18-23. This is perhaps the most novel finding as presented. It would be stronger yet if you could show how it has progressed latitudinally over the decades in line with climate change.

We provide additional analyses to understand the relative individual contribution of autumn and winter, as well as how they compared to the contribution by summer and spring to the annual increasing trend.

We also added a paragraph discussing the relative contribution of the various states along the latitudinal temperature gradient from Tasmania to Queensland.

The discussions are supported by the additional figures:

- Supplementary Figure 2. Seasonal trends of burned area in km². Data: AVHRR-Landgate (1988-2018) and MODIS (2019).
- Supplementary Figure 3. Bi-monthly trends of burned area in km². Data: AVHRR-Landgate (1988-2018) and MODIS (2019).

We further discuss the effect of the temperature latitudinal gradient in the context of the fire return interval in the corresponding section and in Supplementary Table 2.

10. Page 6 Lines 15-18. Good point but what is the corresponding estimate in the amount of cleared forest? Better yet would be the estimate of forest clearing for the period of study.

We removed the mention to the 200-year history of land clearing and make the statement relevant to the last 32-year period of our trend study by citing a recent study of changes in land cover over the same period of our study: Calderon-Loor et al. 2021. Remote Sensing of Environment.

11. Page 6 Lines 23-25. Tell the reader the scale of the “pixel” being referenced here. Added (250m).

12. Figure 2 needs an inset of Australia with an indication of what portion is being shown in

these panels. Not everyone is from Oz. What happened to the forests of Tasmania? Figure 1 of the methods shows a lot more forest there than is apparent in this figure.

We have added an Australia inserts to Figure 2. On Tasmania, the number of years since the last fire was calculated using the pixels that have burned at least once (years since last fire). Please see our response to comment 1 and the expanded information on exactly what we did in Methods: Years since the last fire. We now explain clearly that our approach on Years Since Last Fire (YSLF) is not an attempt to provide a fire frequency for a region or vegetation type, but to use the most robust variable to detect changes in trends. Therefore, as in the case of Tasmania, the YSLF estimates (and pixels in Figure 4) covers the forest that have burned at least once in the 90-year database of burned area from the state and territory agencies (which is only a fraction of the actual forests). Of the total Australian forest cover used in this study of 324,873 km², 170,721 km² had at least one wildfire.

13. Fire return interval – Something akin to Figure 2 but depicting the spatial fire return intervals, as calculated, would be illustrative.

In this ms. we have used Years Since Last Fire (YSLF) based on pixels that have had at least one wildfire. We believe it is the most robust way to detect changes in trends over decades. Unfortunately, we haven't developed a quantitative method that would allow us to produce a fire frequency for regions or vegetation types. The issue is what to assign to pixels that have not burned in the last 90 years because we know that unburned pixels are most certain burned prior to 1930 at an unknown frequency. Previous work has estimated such fire frequencies based on the combination of data and expert solicitation. Our focus on YSLF and our initialization method enables to apply a probability distribution based on the data from each pixel to assert when the last fire occur prior to 1930 so by the time we reach 1930 we can use every pixel year of fire and no fire occurrence as an observation. We only provide the results for the last four decades when there is most cumulative pixel-year information to determine changes in the YSLF.

14. Page 13 Lines 8-10, Hypothesis 2. Fuel loads have increased over time (period). This is due to forest regrowth and vegetative production (NPP). In the absence of sufficient burning these increases will tend to happen regardless of a contribution from potential CO₂ fertilization. CO₂ fertilization effects may have increased rates of increase but nothing here provides context for how much that might be. Jiang et al (2020 Nature) have recently shown that these forests have some of the lowest responsiveness to CO₂ 'fertilization' of any forests globally, so some more context is needed and these hypotheses should be reframed/thought. We agree with the reviewer and we have rephrased the relevant paragraph and the conclusions that comes from it. We no longer link our findings to fire risk, but to show the potential effects of climate change on biomass and fuels. We acknowledge and cite Jian's 2020 paper in the Discussion and Conclusions.

15. General comments. More needs to be done to make clear the assumptions that are being made in making these calculations. 1) all fires are the same (nothing about type (crown, surface) or intensity/severity), 2) all forests are the same (regardless of composition/type, canopy closure, height or age), 3) mortality and fuel consumption are always 100%, resetting fuel age/loading to zero (and all fuels are the same regardless of size, location (surface, canopy, ground), live or dead, or moisture content (e.g. availability)), 4) climate change and weather are even across the continent. Simplifying assumptions are always needed but should be clearly stated, justified to the extent possible, and not be overly simplistic.

We have added more detail on what we have done and the limitations of the data. Particularly for biomass and fuel loads, we state very clearly now that we don't try to directly link

climate, fuel loads and fire risk as we don't have the information required on fuel loads that would enable to do it. The type of information such as fire intensity and severity, or the degree of success in reducing fuel loads from prescribed burning is not available for the continental forest coverage of this study and we are not aware of the modelling tools that could provide enough accurate information. We reflect these limitations with a partially rewritten fuel section and largely rewritten Discussion and conclusions section.

16. Page 23 lines 25-28. It is unclear what to make of this. Litter components should be on the surface of the ground or is this class mixing live foliage and leaf litter? 40% aboveground? Is the other 60% decomposed into humus? The citation given was for fine roots but may have quantified other elements. More clarity needed.

The aboveground biomass fraction includes live material including trunks, branches and leaves, while the litter pool is initially estimated by the model as one combined above and belowground pool, which is decomposed in above and belowground components based on the metadata study by Hendricks et al.1993. We provide these details in Methods: Biomass and fuel loads.

17. Figure 2-SI – needs a corresponding figure of the prescribed (hazardous reduction) burn information.

The trend analyses in this study (and the old Figure 2-SI) are done with the AVHRR-Landgate database given its methodological consistency in space and time (not the case for the State and Territory agencies data). Unfortunately, the AVHRR doesn't detect the low intensity fires which includes most prescribed burns, and therefore an equivalent figure for prescribed burned area cannot be produced. However, we have created a new figure that shows the partition between wildfires and prescribed burning from the States and Territory data for the reader to see the important regional differences (Supplementary Figure 4), in addition to the continental forest Figure 7 in the main ms.

Reviewer #3 (Remarks to the Author):

Overall, this is an interesting, novel and timely analysis relevant to the journal Nature Communications. The analysis is multidimensional in that it explores temporal patterns in increases in forest area burned, along with relevant explanatory mechanisms.

While the analyses of temporal patterns in area burned are sound, the analyses of temporal patterns in fire return interval and some of the explanatory variables are less convincing. For example, the methods for fire return interval do not clearly differentiate between time since fire and interval between fires, with the latter metric being easily the most appropriate dependent variable for analyses. This is compounded by using synthetic data for fuel age whereas both the figures and analyses should be based only on observed intervals between fires.

There are also considerable synthetic elements, in some of the explanatory meteorological and associated variables, which requires further explanation in order to understand the effect of this approach on the study findings.

We respond to these comments in the specific comments below except for one below that do not show later.

Climate and weather surfaces feeding into the various fire weather indices were all taken from the Australian Bureau of Meteorology, which develops and curates the continental datasets for the nation. We have provided some additional references in the ms. It is a very extensive work that we cannot summarize in the available space but hope the additional information will suffice.

As per other variables such as the creation of the database for fire return interval, we have provided more detail on what we did in Methods: Years since last fire, and below in the specific comments.

Finally, the hypothesis relating to CO₂ effects on fuel loads is secondary to a more appropriate hypothesis about what is the combined effect of climate and CO₂ effects on fuel loads over time.

We have removed the hypothesis approach and the specific focus on the elevated CO₂, and focus the discussion on the potential impact of climate change (eg, rainfall and temperature) and the CO₂ fertilization effect, combined and separate (for its mechanistic attribution value).

Further specific comments on aspects of the manuscript:

Page 4, lines 13-16: It is not sufficiently justified why so-called ‘megafire’ years are variously included or excluded in analyses and associated commentary. If subsequent commentary explores the effect of not including 2002 and 2006, then why are they included in the analysis?

Now, all our analyses and regressions include all years although in some instances we present regressions with and without the 2019 fire year to show that increasing trends were already highly statistically significant before the extraordinarily 2019 fire year (supplementary Table 1). We agree with the reviewer that the text could have been misinterpreted as picking years. The Supplementary Table 1 show for some trends regressions with and without the 2019 fire year.

Page 6, lines 15–18: A reference for diminishing extent of forest over the study period, not the period since European invasion, would enhance rigor.

We have added a reference for a recent study on changes in land cover for the same period: Calderon-Loor et al. 2021. *Remote Sensing of Environment*.

Page 7, line 13: The figure caption and associated text refers to times since fire, whereas the section heading is fire return interval. The inclusion in the text, figure and methods (page 21, lines 12 – 13) of time since fire is confusing and somewhat meaningless in this context and the narrative and figures should be based on actual intervals between fires.

We acknowledge that we had used different terminology in an exchangeable manner while the terms indicate different things, which did not help to clearly communicate the intent of the analysis and the rationale for our choices. We have now tightened the use of terminology and explain and justify further what we did in Methods and here.

We agree that the most important quantity is fire frequency requiring pixels that at least had two fires. Given the fire frequency of these forests is anywhere between 20 and more than 100 years, our 90-year record falls short for many fire regimes to capture two or more fires and provide a pixel population large enough to show robust statistics. We also realize that narrowing the population of pixels so much with at least two fires, would also result in fire frequencies much higher than would not be relevant to the fire regimes of the regions or vegetation classes. For these reasons, we think that years since the last wildfire (YSLF) provides some advantage to detect trends:

- Most statistically robust approached as it uses a much larger pixel-year population than if we do return intervals.
- The YSLF gives the current state of wildfire fire regime, whereas mean fire return interval lags one fire event behind. That is, mean FRI will be lagged wrt to mean YSLF. Because our focus of this paper is to detect changes in trends in fire activity, we think that although not the ideal quantity to from an ecosystem perspective and fire regimes, the YSLF is more powerful in detecting changes in trends all the way to the 2019 fire year.
- YSLF is easy to initialise, and gives pretty much real data (observations on fire and no-fire) from the beginning of the timeseries (see more details in response to Page 21, lines 22–30 in the way we initialize the dataset), while calculating FRI always requires 2 events.
- We only focus on years since the last wildfire, which excludes prescribed burning and therefore our focus is not on fuel age. The purpose of the analysis is to understand changes in fire activity that could be related to changes in climate, and therefore requiring to remove any direct human impacts such as the frequency and quantity of prescribed burning.

Page 8, line 6: McArthur’s FFDI indicates weather conditions across the ‘Danger’ spectrum, not just dangerous weather conditions.

We changed “dangerous weather conditions” with “weather conditions across the “Danger” spectrum”.

Page 8, line 8: McArthur’s FFDI incorporates a drought index and a ‘Drought Factor’, not a fuel moisture deficit.

We have revised the wording and use Drought Factor.

Page 13, lines 9-10: There is another hypothesis regarding climate change effects on fuel loads, being that fuel loads are declining due to decreased vegetative productivity associated with drier conditions.

We agree with the reviewer, and it is very possible that this is the case in some regions. Our coarse modelling show that aggregated across all forests, the potential CO₂ fertilization effect is still wining over the negative effects of increase temperature and rainfall decline. However, a key fuel type, fine fuels, showed a declining trend in response to climate change. We have revised the text to make clear that this is a modelling exploration of the possible impacts of climate change and the CO₂ fertilization and we do not link it directly to changes in fire risk given the limitations of the data.

‘Figure 4’ appears twice and Figure 5 caption is missing.

Thanks, it was a misnumbering of the figures. Corrected.

Page 14, line 2: ‘prescribed burnings’ should be written as ‘prescribed burning’. Same for page 19, line 19.

Corrected throughout the ms.

Figure 4, the second one: There does not appear to be any orange-coded data indicated. If it is too limited in area to be visible in the figure then that should be indicated in the figure caption.

Yes, the amount of prescribed burning is so small in the ACT that cannot be seen in the figure. We have now merged NSW with ACT in one single color. The merge also makes

sense from a geographical point of view.

Page 15, lines 6–10: Overall, if fine litter is projected to decline (Fig 6b), it is largely irrelevant for this study if the CO₂ effect in isolation is to increase aboveground biomass. We have now rewritten some parts of the biomass section and we no longer make CO₂ fertilization effect the main focus, but both climate change and CO₂. We agree that what is important is the net result of all effects.

Page 16, line 15 to page 17, line 2: A paragraph of the conclusion is effectively repeated. Second paragraph deleted.

Some hyperlinks in the document are broken, e.g. Page 20, lines 30–31. All links in the ms. now working.

Page 21, lines 2–3: The sentence does not make sense. Fixed.

Page 21, lines 22–30: The figures and analyses should be limited to realized fire intervals, without including any aspect of synthetic fuel age. Given the clarification on page 22, line 2 then the method should just state the analysis is based on observations, without invoking the synthetic estimates of fuel age in the manuscript. We have provided additional information in Methods on what we did, and show that the “synthetic” data is an approach to be able to use more fully the observations in the State and Territory agency datasets, not to dilute the robustness of the actual observations. The initialization method ensures that we use every single pixel-year data point from 1930 onwards, either as a fire or no-fire observation. That is, if a particular pixel first burned in eg, 1980 and a second one in 2006, by only using the second fire to determine a fire return interval, we would have 50 years of no-fire observations (1930-1979) not contributing to the analysis. The initialization method applies a probability distribution based on the data from each pixel to assert when the last fire prior to 1930 occurred. This enable that by the time we reach 1930 we can use every pixel-year of fire and no fire occurrence as an observation. Once the first fire occurs in a pixel, then the calculations proceed based on the changes in the YSLF in that very pixel. Because we only provide YSLF results for the last four decades, the mean values of those decades are little or no effected by the initialization other than having a much larger pixel-year population onto which based the calculations. This approach is an attempt to overcome the limitations of a 90-year long dataset which is still too short to estimate fire regims with fire return intervas between 20 and more than 100 years. See Methods with the additional information we provided.

Page 23, lines 19–22: Again, the reliance on synthetic data is not convincing for this application.

Please refer to the explanations above. We hope we have better explained in the methods and above what we did.

Table 1 – S1: The table needs to be carefully checked. There are several repeated pairs of dependent and independent variables (e.g. numbers 7 and 9; numbers 8 and 10). We have checked, fixed and expanded the table. Please notice that we have some of the entries that are repeated with and without 2019 fire season to show that trends were highly significant and not over influenced by the extraordinary high burned area of the 2019 fire year. We have also expanded the regression analysis with new text in section “Burned area

and fire weather” with all parameters of the regressions in Supplementary Table 3.

REVIEWER COMMENTS

Reviewer #1 (Remarks to the Author):

Re-review: NCOMMS-20-29739A - Multi-decadal increase of forest burned area in Australia is linked to climate change

The entire document is well written and accessible to all readers to understand some complex components of fire and climate.

The authors have addressed most of my reviewer comments and conducted further work to alleviate concerns.

The introduction provides clear background and rationale for the study and covers all the main points although biomass production could be expanded to prepare readers for the fire risk and fuel load section and related uncertainties.

The clarification and additional figures describing the different fire datasets are helpful and improve the manuscript.

The addition of the multivariate analysis and results (starting page 13) are useful but readers will similarly wonder if there is spatial variability in these results.

However, I still have the same concerns about the Fire Risk Factors and fuel load section (starting page 15).

I still find this section disjointed from the rest of the paper. This section outlines the importance of fuel load to fire risk, summarises the lack of a trend in prescribed burning and finds, due to changes in CO₂, a small trend in aboveground biomass and coarse woody debris but a declining trend in fine litter. However, none of this is linked back to the burned area analyses which is a focus of the paper. Is there a relationship between the modelled biomass changes and burned area?

Additionally, I think it is worth ensuring the paper articulates that fuel structure, continuity and conditions are indicators of fire risk rather than fuel load.

Finally, in the discussion and conclusion section there is a statement that it is very likely that fuel management had no effect on the increasing trend in burned areas but it must be noted that the objective of fuel hazard reduction burning is not to reduce area burned so there should be no expectation that fuel management would have an effect on burned areas.

Minor comments:

A paper published recently in Nature by Abrams et al 2021 Connections of climate change and variability to large and extreme forest fires in southeast Australia

<https://www.nature.com/articles/s43247-020-00065-8> covers some similar topics. It might be worth the authors exploring if there are any overlaps that need to be identified. Noting that one of the authors (Andrew Dowdy) is listed on both papers so I assume they are across the detail of both.

Page 2, Lines 5-7: need a term at the end of the sentence that is all encompassing – it currently doesn't make sense.

Page 5, Line 18: I am impressed/amazed at the alignment between agency fire records and satellite detected burnt areas. I think the addition of this plot is very worthwhile not just for the study but for other pieces of work.

Page 7, line 4: change 'percintile' to 'percentile'

Page 16, line 18: change 'cool' to 'cooler'

Page 16, line 21: might be worth clarifying/referencing the 'increasing hunting productivity'. Do the authors mean that hunting productivity is improved for animals or humans?

Page 20, line 17-20: this statement that it is very likely that fuel management had no effect on the increasing trend in burned areas but it must be noted that the objective of fuel hazard reduction burning is not to reduce area burned.

Reviewer #2 (Remarks to the Author):

This is a review of the revised manuscript. The authors' investigated changes in annual burned area in forests of Australia using both a satellite-derived (AVHRR) burned area product (32 years) and integrated State and Territorial data (90 years). They conclude that the trend in burned area over time is increasing significantly and that the most likely cause is changing climate conditions. The manuscript is very well written, per se, but could have used some proof reading with a spellchecker. Overall, the manuscript is much improved with only a few remaining issues.

Specific points follow.

1. Page 1 line 23 – “burnd”

2. Figure 1 – While it is clear why 2019 had to be a MODIS point versus the AVHRR used since the 1980s, the data for all years of the MODIS record should be plotted here together with the AVHRR. Doing so would provide much better context for the believability of the 2019 data point. Given the stated correspondence between AVHRR and MODIS in other years this should strengthen the authors' arguments. As things stand the MODIS point looks potentially erroneous. The authors do a good job in the text giving results with and without the 2019 point, which helps but just plotting the MODIS data would help a lot. Note Figure S1 shows the MODIS and AVHRR are reasonably close so plotting those data here would be strongly advised.

3. Page 5 lines 26-27. The other seasons contribute more of the increase in area. Are the trends significant for the seasons besides Autumn?

4. Figure 2 – This is a very nice figure once examined for a bit. I am not sure “heat map” is accurate since it is of ‘area’ but the intent is clear and I think the terminology is used elsewhere. That said, the inferences apparent in the heat map are heavily weighted by 2019 which is from a different sensor (MODIS) than the 1987-2018 (AVHRR) portion of the figure. This is another case where having plotted the full MODIS record in Figure 1 might give the reader confidence in the 2019 line of data here. Without some way of assessing the correspondence of the two sets of imagery, having the 2019 plotted here in this way is potentially misleading. There is a good chance that MODIS will detect less area than AVHRR so this figure may be understated but the reader has no way of knowing this. Although it mentions that 2019 is from MODIS in the caption it would have to be highlighted in some way (akin to the triangle of Figure 1 or circled dot in Figure 3) or separated in an offset panel, ideally showing the corresponding MODIS time series data adjacent to the AVHRR series. Additionally, the figure should clearly indicate which months are the Autumn/Winter versus the Spring/Summer so the reader can more easily understand your reported findings about increases in the burning season.

5. Page 7 line 3 – since ‘the’ 1930s

6. Page 7 line 4 – “percintile”

7. Note on the Year Since Last Fire (YSLF) calculations – While it would be better to put all of this in terms of the actual forested area, regardless of whether or not it has burned since 1930 (pre-1930 burned area could just be referenced as >90 years since burning), I can see how this approach could be used. My concern was that the calculations were being made incrementally year by year such that the total forested area under consideration was increasing each year from any new first-time burned areas of the given year. In reading through this version, I believe what it is saying is that the “forest” area used for the calculations was the sum of all areas that are known to have burned at least once since 1930. Just make a clear and explicit statement to that effect in the main text and a lot of confusion can be avoided.

8. Page 9 Figure 4 – What is the spatial grain size of this map? What is actually meant here by “decadal mean”? Is this a spatial mean aggregating of YSLF values of the component pixels? If so, at what scale? If it means something else then it needs better definition/explanation. How does this figure differ from a simple YSLF map?

9. Page 11 Line 30 – content ‘that’ is as expected
10. Page 12 FFDI and lightning effects on burned area – How does this differ from the detailed work up of these associations and data by Williamson et al (2016 ERL)?
11. Figure 5 – Tx is not defined anywhere and would probably be better stated as Tmax as it is most everywhere else.
12. Figure 6 – helpful to have small inset text for each of the regressions with R2 and p-values for each panel.
13. Page 17 Line 8 – The underlying cause of climate change is much more than just CO2. CO2 makes up ~60% of the added climate forcing. It is, however, the portion important for this analysis. The easiest thing would be to just reword here as “Climate change and the associated accumulation of anthropogenic CO2 emissions....”
14. Page 17 Line 14 – delete “including”
15. Page 19 Lines 12-18 – FYI, Jolly et al. (2015 Nature Communications) showed that the fire weather trends and particularly the increasing fire season length into the cooler months were most pronounced at the northern and southern extremities of eastern Australia during recent decades – exactly where you are reporting fire activity increasing in this manuscript. Combined, that’s rather compelling.
16. Page 19 Line 22 – “rapid shortening of the fire return interval in forest ecosystems across Australia” -- The rebuttal letter item #1 states that in sidestepping previous criticism of the improper use of terminology that the manuscript now has “the exclusive use of “years since the last fire, YSLF”” and yet at this point fire return interval is used here again interchangeably with YSLF which it is definitely not, at least as used in the related fire literature. Correct this or actually work up fire return interval information. This could be done if framed and analyzed correctly.
17. Page 19 Lines 25-26 – “however, pixel level analysis reveals that mean decadal values of YSLF in some areas are already outside historical fire regimes (Figure 4)” appears to be a nonsensical statement. By definition all areas that burned within a given decade burned within 10 years and so would be less than the time period of your cited most vulnerable species that need 20-30 years to reproduce seed. To make this point weakly you’d need to be looking at the time period between 2 known fires at a given location. To really make it though you’d want to relate that period to the proportion of the landscape with the specific forest type that has experienced this frequent burning. Note, having a few pixels worth that burn more frequently than the typical fire return interval is to be expected, as is having a portion burning more infrequently. It can be said that more area is burning in recent decades and that if that trend continues, especially in areas with sensitive species that it could prove ecologically problematic but nothing here seems to be doing that and there is no spatial information on forest types that can be related to the fire data being presented.
18. Page 20 Line 1 – “mortality”
19. Page 20 Lines 27-30 – Get the tenses of this sentence sorted out.
20. Page 28 Lines 21-22. – Averaged how? The “methods” don’t help regarding this at all. What is averaged over what period of space and time? Model is at ~25km, Forest map is at 1.1 km (AVHRR), or 250m for fire. How do fires during the time period being examined impact averages?
21. Page 28 line 28 – TRENDY protocols are mentioned here and nowhere else. What is it?
22. References number 11 – “Brastock” should be Bradstock
23. Page 43 Figure S7 – Which dots are the 2019 MODIS points? The other figures are careful to show this clearly.

Reviewer #3 (Remarks to the Author):

The authors' writing in their review response is very poor quality in places, limiting the extent to which authors' responses can be properly evaluated. Authors may share the view that good quality correspondence is not required in responding to reviews. However, that view is not shared by this reviewer. Many of these authors have previously demonstrated very good writing capacity, although that characteristic is not evident in key aspects of their review response. Examples of poor communication are listed below.

"... as the model does not reset loads after disturbances and show no correlation because are potential quantities (supplementary Figure 7)."

"However, we only provide results for the past four decades (to overlap with the AVHRR record) which the data is likely to be most complete."

"It shows a remarkable consistent of the main features of the multi-decadal trends, ..."

"We now provide more discussion and figures to unrevealed what is contributing to ..."

"We realize we used the various terms of fire frequency, fuel age and fire return interval a bit lose leading to confusion."

"The focus on YSLF for cells with fire occurrence data enabled to do this study most quantitatively and robustly, ..."

"We have rewritten the conclusions from the biomass analysis to be more nuance and ..."

"Therefore, we are carefully to say ..."

"Our focus on YSLF and our initialization method enables to apply a probability distribution based on the data from each pixel to assert when the last fire occur prior to 1930 so by the time we reach 1930 we can use every pixel year of fire and no fire occurrence as an observation."

"Most statistically robust approached as it uses a much larger pixel-year population than if we do return intervals."

"... we think that although not the ideal quantity to from an ecosystem perspective ..."

"... the potential CO2 fertilization effect is still winning over ..."

"This enable that by the time we reach ..."

REVIEWER COMMENTS

Reviewer #1 (Remarks to the Author):

Re-review: NCOMMS-20-29739A - Multi-decadal increase of forest burned area in Australia is linked to climate change

The entire document is well written and accessible to all readers to understand some complex components of fire and climate.

The authors have addressed most of my reviewer comments and conducted further work to alleviate concerns.

The introduction provides clear background and rationale for the study and covers all the main points although biomass production could be expanded to prepare readers for the fire risk and fuel load section and related uncertainties.

The clarification and additional figures describing the different fire datasets are helpful and improve the manuscript.

The addition of the multivariate analysis and results (starting page 13) are useful but readers will similarly wonder if there is spatial variability in these results.

However, I still have the same concerns about the Fire Risk Factors and fuel load section (starting page 15).

I still find this section disjointed from the rest of the paper. This section outlines the importance of fuel load to fire risk, summarises the lack of a trend in prescribed burning and finds, due to changes in CO₂, a small trend in aboveground biomass and coarse woody debris but a declining trend in fine litter. However, none of this is linked back to the burned area analyses which is a focus of the paper. Is there a relationship between the modelled biomass changes and burned area?

Figure S7 (below) shows the relationships between burned area and biomass, with no significant trends. We make this explicit in the text when discussing the multivariate analysis and fuel loads. However, we want to emphasize that the simulated biomass over time with the CABLE model is an estimate of the potential biomass trends given observed changes in the climate and atmospheric CO₂. It is not a simulation of actual biomass and fuel loads because the model does not track previous fires and other disturbances, which reduces fuel loads when compared to the non-disturbance state that we are able to model with CABLE. Because of that, we are more cautious in drawing conclusions from the analysis about the role of biomass on burned area trends than we are for the climate and weather variables for which we rely on many observations. This might be the reason why the reviewer finds our treatment of the biomass component not as complete and integrated as the climate component. This is an exclusive reflection of the data available and what we can draw from it by way of conclusive statements. We had explored whether datasets on historical fuel loads covering the area of interest existed and found none; an important research area for the future.

Additionally, I think it is worth ensuring the papers articulates that fuel structure, continuity and conditions are indicators of fire risk rather than fuel load.

We have rewritten the relevant paragraph which now includes: “In addition to weather and climatic conditions, and the availability of ignition sources for a fire to occur, fuel amount, structure, continuity and condition are also key components of fire risk”.

Finally, in the discussion and conclusion section there is a statement that it is very likely that fuel management had no effect on the increasing trend in burned areas but it must be noted that the objective of fuel hazard reduction burning is not to reduce area burned so there should be no expectation that fuel management would have an effect on burned areas.

We have added the following sentence to make clear the role of fuel management and to avoid possible misinterpretations from our results.

“We also note that the main objective of fuel management is to reduce fire risk and severity (Morgan et al. 2020, Prescribed burning in south-eastern Australia: history and future directions. Forestry Australia), which might or might not result in reduced total burned area”

Minor comments:

A paper published recently in Nature by Abrams et al 2021 Connections of climate change and variability to large and extreme forest fires in southeast

Australia <https://www.nature.com/articles/s43247-020-00065-8> covers some similar topics.

It might be worth the authors exploring if there are any overlaps that need to be identified.

Noting that one of the authors (Andrew Dowdy) is listed on both papers so I assume they are across the detail of both.

Yes, the paper is important and covers fire weather trends, how they are influenced by the compounded effects of climate modes in Australia, and how unique the 2019 fire season was in the long-term paleo context. The paper is highly complementary to the main focus of our analysis on burned area and trends, while we do not focus on what drove the 2019 fire season nor the underlying climate modes driving fire weather. We cite the paper in our ms. to highlight the prior knowledge on major climate drivers of fire weather trends and variability.

Page 2, Lines 5-7: need a term at the end of the sentence that is all encompassing – it currently doesn't make sense.

Corrected.

Page 5, Line 18: I am impressed/amazed at the alignment between agency fire records and satellite detected burnt areas. I think the addition of this plot is very worthwhile not just for the study but for other pieces of work.

Thanks.

Page 7, line 4: change 'percintile' to 'percentile'

Corrected.

Page 16, line 18: change 'cool' to 'cooler'

Changed.

Page 16, line 21: might be worth clarifying/referencing the 'increasing hunting productivity'. Do the authors mean that hunting productivity is improved for animals or humans?

We meant increasing hunting for first nations Aboriginal communities. We have rephrased as: "...reducing fire intensity and increasing hunting for first nations communities"

Page 20, line17-20: this statement that it is very likely that fuel management had no effect on the increasing trend in burned areas but it must be noted that the objective of fuel hazard reduction burning is not to reduce area burned.

As referred in a previous comment, we have added: "We also note that the main objective of fuel management is to reduce fire risk and severity (Morgan et al. 2020, Prescribed burning in south-eastern Australia: history and future directions. Forestry Australia), which might or might not result in reduced total burned area"

Reviewer #2 (Remarks to the Author):

This is a review of the revised manuscript. The authors' investigated changes in annual burned area in forests of Australia using both a satellite-derived (AVHRR) burned area product (32 years) and integrated State and Territorial data (90 years). They conclude that the trend in burned area over time is increasing significantly and that the most likely cause is changing climate conditions. The manuscript is very well written, per se, but could have used

some proof reading with a spellchecker. Overall, the manuscript is much improved with only a few remaining issues.

Specific points follow.

1. Page 1 line 23 – “burnd”

Corrected.

2. Figure 1 – While it is clear why 2019 had to be a MODIS point versus the AVHRR used since the 1980s, the data for all years of the MODIS record should be plotted here together with the AVHRR. Doing so would provide much better context for the believability of the 2019 data point. Given the stated correspondence between AVHRR and MODIS in other years this should strengthen the authors’ arguments. As things stand the MODIS point looks potentially erroneous. The authors do a good job in the text giving results with and without the 2019 point, which helps but just plotting the MODIS data would help a lot. Note Figure S1 shows the MODIS and AVHRR are reasonably close so plotting those data here would be strongly advised.

In response to this review comment we made two major improvements to the manuscript and Figure 1:

- 1) Given the time passed since the last review, we have now been able to acquire the AVHRR data for the 2019 fire season and we no longer need to complete the AVHRR series with MODIS for the 2019 season. We have permeated this change to all figures and trend analyses in the manuscript.
- 2) We have added the shorter complete MODIS shorter series in Figure 1 for comparison as requested.

We also want to highlight that Figure S1 compares the annual burned area for the three datasets reinforcing the overall trends.

3. Page 5 lines 26-27. The other seasons contribute more of the increase in area. Are the trends significant for the seasons besides Autumn?

All seasons show highly significant trends ($p\text{-value} < 0.001$) and $p\text{-value} = 0.07$ for the summer season. We provide this information in the text and caption of Supplementary Figure 2.

4. Figure 2 – This is a very nice figure once examined for a bit. I am not sure “heat map” is accurate since it is of ‘area’ but the intent is clear and I think the terminology is used elsewhere. That said, the inferences apparent in the heat map are heavily weighted by 2019 which is from a different sensor (MODIS) than the 1987-2018 (AVHRR) portion of the figure. This is another case where having plotted the full MODIS record in Figure 1 might give the reader confidence in the 2019 line of data here. Without some way of assessing the correspondence of the two sets of imagery, having the 2019 plotted here in this way is potentially misleading. There is a good chance that MODIS will detect less area than AVHRR so this figure may be understated but the reader has no way of knowing this. Although it mentions that 2019 is from MODIS in the caption it would have to be highlighted in some way (akin to the triangle of Figure 1 or circled dot in Figure 3) or separated in an offset panel, ideally showing the corresponding MODIS time series data adjacent to the AVHRR series. Additionally, the figure should clearly indicate which months are the

Autumn/Winter versus the Spring/Summer so the reader can more easily understand your reported findings about increases in the burning season.

We found that both the literature and graphic packages called the figure a “heat map”. However, given the fire topic of the ms., we appreciate that the reader might initially think the figure is about temperature. We have now removed any mention of “heat map” in the manuscript.

With regard to using MODIS for the 2019 fire season, we have now replotted the figure using the newly available AVHRR-Landgate series all the way to the 2019 fire season; we no longer need to use MODIS to complete the series. It is worth noting that of the three estimates of the area burned for the 2019 fire season presented in our study, MODIS gives the smallest area, but all three estimates are highly consistent in estimating an unprecedented extent of burned area. See Figure 3 and also have added the text:

“The 2019 fire year burned about three times ($60,345 \text{ km}^2$) the area of any previous year in the 32-year AVHRR-Landgate record (Figure 3, Supplementary Fig. 1, Methods: Burned area). The burned area of the 2019 fire year was estimated at $71,772 \text{ km}^2$ based on State and Territory agencies (NIAFED) and $54,852 \text{ km}^2$ based on NASA-MODIS, with an average for the three products of $62,323 \pm 8,631$.”

We have also named the four seasons in figure 2 to ensure the reader does not mistake them for Northern Hemisphere seasons.

5. Page 7 line 3 – since ‘the’ 1930s

Added.

6. Page 7 line 4 – “percintile”

Corrected.

7. Note on the Year Since Last Fire (YSLF) calculations – While it would be better to put all of this in terms of the actual forested area, regardless of whether or not it has burned since 1930 (pre-1930 burned area could just be referenced as >90 years since burning), I can see how this approach could be used. My concern was that the calculations were being made incrementally year by year such that the total forested area under consideration was increasing each year from any new first-time burned areas of the given year. In reading through this version, I believe what it is saying is that the “forest” area used for the calculations was the sum of all areas that are known to have burned at least once since 1930. Just make a clear and explicit statement to that effect in the main text and a lot of confusion can be avoided.

Yes, the reviewer is correct that we are using the same area of forests throughout the four decades (and from 1930 when the calculations begin), that is, the area of forests that has burned at least once since records began. We have made that clearer in the main text.

To provide more information for the reader to understand what is covered in the analysis and make sure that YSLF is not mistaken for a fire frequency for the entire landscape, we have added the following text:

“The analysis shows that 48.8% of all forest area has burned at least once since the 1930s. The burned fraction of the total area for each of the forest classes consistent with the Australian National Forest Inventory (Methods: Forest extent and types) varied widely: eucalypt low-forest 61.1%, eucalypt tall-forest 60.6%, eucalypt medium-forest 58.2%), coastal non-eucalypt 54.3%, and rainforest 11.1%).”

8. Page 9 Figure 4 – What is the spatial grain size of this map? What is actually meant here by “decadal mean”? Is this a spatial mean aggregating of YSLF values of the component pixels? If so, at what scale? If it means something else then it needs better definition/explanation. How does this figure differ from a simple YSLF map?

The map shows the forest area which has burned at least once since records began in each of the states and territories, for most states around the 1930s. It is at 0.0025-degree (approximate 250 m) resolution rasterised from the original vector-mapped fire history. Each panel of the figure is a map of mean decadal YSLF; each pixel’s value in the panel is the average of the 10 annual pixel values that occurred at the grid cell location during the specified decade. It is not a spatial mean; it is the (temporal) mean of the 10 YSLF grids that comprise the decade. We have made it clearer in the legend of Figure 4.

9. Page 11 Line 30 – content ‘that’ is as expected

Added.

10. Page 12 FFDI and lightning effects on burned area – How does this differ from the detailed work up of these associations and data by Williamson et al (2016 ERL)?

The paper by Williamson et al. (2016) is in broad agreement with many of the points from our study and others on fire weather and lightning (including for most of the conclusions presented in their Abstract), while noting a key point of difference around trend identification for FFDI. In particular, Williamson et al. conclude that "Our study demonstrates that Australia has a long fire weather season with high inter-annual variability relative to all other continents, making it difficult to detect long term trends." We agree that the high inter-annual variability makes trend difficult to determine. However, the FFDI dataset we used in this study clearly shows a significant long-term trend, as has been demonstrated previously in other papers (Dowdy 2018; Abram et al. 2021), noting that this dataset covers a longer time period of about 70 years than the data used in Williamson et al. (2016). The FFDI dataset we used is based on a gridded analysis of observation data which can help provide insights on regional detail and variations through Australia. The Williamson et al. (2016) paper based on different data and methods is therefore important in providing a complementary perspective to these other recent studies (Dowdy 2018; Harris and Lucas 2019; Abram et al. 2021). We now cite the paper in the ms.

11. Figure 5 – Tx is not defined anywhere and would probably be better stated as Tmax as it is most everywhere else.

We replaced Tx with Tmax in the revised figure.

12. Figure 6 – helpful to have small inset text for each of the regressions with R2 and p-values for each panel.

Added.

13. Page 17 Line 8 – The underlying cause of climate change is much more than just CO2. CO2 makes up ~60% of the added climate forcing. It is, however, the portion important for this analysis. The easiest thing would be to just reword here as “Climate change and the associated accumulation of anthropogenic CO2 emissions....”

Changed, thank you for the text suggestion.

14. Page 17 Line 14 – delete “including”

Deleted.

15. Page 19 Lines 12-18 – FYI, Jolly et al. (2015 Nature Communications) showed that the fire weather trends and particularly the increasing fire season length into the cooler months were most pronounced at the northern and southern extremities of eastern Australia during recent decades – exactly where you are reporting fire activity increasing in this manuscript. Combined, that’s rather compelling.

Thank you for making the link to a previously published paper. We notice in this global analysis that Australia is behaving quite different from the rest of the world too.

16. Page 19 Line 22 – “rapid shortening of the fire return interval in forest ecosystems across Australia” -- The rebuttal letter item #1 states that in sidestepping previous criticism of the improper use of terminology that the manuscript now has “the exclusive use of “years since the last fire, YSLF”” and yet at this point fire return interval is used here again interchangeably with YSLF which it is definitely not, at least as used in the related fire literature. Correct this or actually work up fire return interval information. This could be done if framed and analyzed correctly.

We have replaced “fire return interval” with “fire since last year”.

17. Page 19 Lines 25-26 – “however, pixel level analysis reveals that mean decadal values of YSLF in some areas are already outside historical fire regimes (Figure 4)” appears to be a nonsensical statement. By definition all areas that burned within a given decade burned within 10 years and so would be less than the time period of your cited most vulnerable species that need 20-30 years to reproduce seed. To make this point weakly you’d need to be looking at the time period between 2 known fires at a given location. To really make it though you’d want to relate that period to the proportion of the landscape with the specific forest type that has experienced this frequent burning. Note, having a few pixels worth that burn more frequently than the typical fire return interval is to be expected, as is having a portion burning more infrequently. It can be said that more area is burning in recent decades and that if that trend continues, especially in areas with sensitive species that it could prove ecologically problematic but nothing here seems to be doing that and there is no spatial information on forest types that can be related to the fire data being presented.

Thanks for pointing out that the sentence didn't convey the intended message. We have now rewritten it. What we mean is that there is a growing trend over time of the number of pixels (the basis of our analysis) for which the years since the last fire is decreasing, particularly for some regions such as Victoria.

The decadal mean for a given pixel is the mean of its age-since-fire when the decade starts and when the decade ends. If the pixel doesn't burn during the decade, its mean YSLF for the decade will be its age-since-fire upon entering the decade (t_{init}) plus 5. If it burns during the decade, its age is reset to zero. In that case, the pixel's decadal mean YSLF will be substantially less than $t_{init}+5$ if the burn happens early in the decade, but only somewhat less than $t_{init}+5$ if the burn happens late in the decade, in which case the effect of the burn on YSLF at that pixel will be mostly realised in the following decade.

We agree that we can only make statements that relate to the increasing amount of area under decreasing YSLF and that it is for now just a local phenomenon, because we do not provide the region-wide numbers of each vegetation type. We also hope that the information now provided on the fraction of each major forest type that has burned at least once, and therefore part of the analyses (response to a comment above), will help the reader to understand what we have done with a focus on trends, not in defining a landscape-wide fire regime.

The new sentence reads:

"...,however, we find a growing trend in the number of pixels in which YSLF is decreasing (Figure 4). A continuation of that trend, especially in areas with sensitive species, could lead to significant ecological changes. Examples of sensitive ecosystems are the mountain ash (*Eucalyptus regnans*) and alpine ash (*Eucalyptus delegatensis*) forests in Victoria"

18. Page 20 Line 1 – "mortality"

Corrected.

19. Page 20 Lines 27-30 – Get the tenses of this sentence sorted out.

Fixed.

20. Page 28 Lines 21-22. – Averaged how? The "methods" don't help regarding this at all. What is averaged over what period of space and time? Model is at ~25km, Forest map is at 1.1 km (AVHRR), or 250m for fire. How do fires during the time period being examined impact averages?

All biomass and fuel stocks were averaged spatially across all grid cells dominated by forest ecosystems and for each year. We have provided the additional information in the new sentence below:

"Biomass (including wood and leaf components) and litter stocks (coarse woody debris, fine litter, and very fine litter) (all in kg C m^{-2}) were averaged spatially across all grid cells dominated by forest ecosystems and for each year"

CABLE does not track previous fires. We make clear in the main text that the simulated changes in biomass are the changes due to changes of climate and atmospheric CO_2 , but do not include the effect of disturbances (such as previous fires), which we know are important for fuel loads. In addition, we have added the following sentences in Methods: "CABLE

provides potential biomass and fuel loads based on climate and CO₂ and their changes over time, but does not simulate previous fires which also affect fuel loads.”

21. Page 28 line 28 – TRENDY protocols are mentioned here and nowhere else. What is it?

We have provided additional explanations and provide a link where the protocol is fully described

“TRENDY (trends in the land carbon cycle) is a model intercomparison of state-of-the-art Dynamic Global Vegetation Models (DGVMs) with common model experiments and driving datasets. It is used to estimate the global land sink for the Global Carbon Project’s annual update of the global carbon budget⁵³. The protocol is fully described here <https://sites.exeter.ac.uk/trendy/>.”

22. References number 11 – “Brastock” should be Bradstock

Corrected.

23. Page 43 Figure S7 – Which dots are the 2019 MODIS points? The other figures are careful to show this clearly.

We have added a triangle in Figure S7 to show the fire season of 2019. However, we are pleased to say that given the time passed since the last review, we have now been able to acquire the AVHRR-Landgate data for the 2019 fire season (to June 2020). Therefore, we no longer need to complete the AVHRR series with MODIS for the 2019 season. We have permeated this change to all figures and trend analyses in the manuscript.

However, given the value of having an additional dataset to build robustness into the numbers and trends, we now show two or three datasets as relevant in figures 1, 3, S1.

Reviewer #3 (Remarks to the Author):

The authors’ writing in their review response is very poor quality in places, limiting the extent to which authors’ responses can be properly evaluated. Authors may share the view that good quality correspondence is not required in responding to reviews. However, that view is not shared by this reviewer. Many of these authors have previously demonstrated very good writing capacity, although that characteristic is not evident in key aspects of their review response. Examples of poor communication are listed below.

“... as the model does not reset loads after disturbances and show no correlation because are potential quantities (supplementary Figure 7).”

“However, we only provide results for the past four decades (to overlap with the AVHRR record) which the data is likely to be most complete.”

“It shows a remarkable consistent of the main features of the multi-decadal trends, ...”

“We now provide more discussion and figures to unrevealed what is contributing to ...”

“We realize we used the various terms of fire frequency, fuel age and fire return interval a bit loose leading to confusion.”

“The focus on YSLF for cells with fire occurrence data enabled to do this study most quantitatively and robustly, ...”

“We have rewritten the conclusions from the biomass analysis to be more nuance and ...”

“Therefore, we are carefully to say ...”

“Our focus on YSLF and our initialization method enables to apply a probability distribution based on the data from each pixel to assert when the last fire occur prior to 1930 so by the time we reach 1930 we can use every pixel year of fire and no fire occurrence as an observation.”

“Most statistically robust approached as it uses a much larger pixel-year population than if we do return intervals.”

“... we think that although not the ideal quantity to from an ecosystem perspective ...”

“ ... the potential CO2 fertilization effect is still wining over ...”

“This enable that by the time we reach ...”

We are very sorry that we submitted a document that was not fully checked for spelling mistakes and syntax. The mistakes were caused by not realizing the spelling tool was not working, and it did not represent the time and care we put in improving the ms. and responding to the reviewers.

We have carefully revised the ms., addressed your comments, and revised the responses below.

Overall, this is an interesting, novel and timely analysis relevant to the journal Nature Communications. The analysis is multidimensional in that it explores temporal patterns in increases in forest area burned, along with relevant explanatory mechanisms.

While the analyses of temporal patterns in area burned are sound, the analyses of temporal patterns in fire return interval and some of the explanatory variables are less convincing. For example, the methods for fire return interval do not clearly differentiate between time since fire and interval between fires, with the latter metric being easily the most appropriate dependent variable for analyses. This is compounded by using synthetic data for fuel age whereas both the figures and analyses should be based only on observed intervals between fires.

There are also considerable synthetic elements, in some of the explanatory meteorological and associated variables, which requires further explanation in order to understand the effect of this approach on the study findings.

The various points raised in this comment are addressed individually below in response to each specific comment, except for the following:

Climate and weather surfaces feeding into the various fire weather indices were taken from the Australian Bureau of Meteorology, which develops and curates the continental datasets for the nation. We have provided additional information and references in the ms. to trace back the methods, which have not been developed in our study. We hope that this will help the reader to more clearly explore the multiple sources of methods and data.

Finally, the hypothesis relating to CO₂ effects on fuel loads is secondary to a more appropriate hypothesis about what is the combined effect of climate and CO₂ effects on fuel loads over time.

We have removed the hypothesis and the specific focus on the elevated CO₂. We now focus the discussion on the potential impact of climate change (eg, rainfall and temperature) and the CO₂ fertilization effect, combined and separately. We agree that the most important result is the combined effects of climate and CO₂, and we have only left the individual effects in the ms. for their value in understanding the underlying mechanisms driving the changes.

Further specific comments on aspects of the manuscript:

Page 4, lines 13-16: It is not sufficiently justified why so-called ‘megafire’ years are variously included or excluded in analyses and associated commentary. If subsequent commentary explores the effect of not including 2002 and 2006, then why are they included in the analysis?

All our analyses and regressions now include all years although in some instances we present regressions with and without the 2019 fire year to show that increasing trends were already highly statistically significant before the extraordinary 2019 fire year (Supplementary Table 1). We agree with the reviewer that the choices made before could have been misinterpreted as picking years. This is now fixed.

Page 6, lines 15–18: A reference for diminishing extent of forest over the study period, not the period since European invasion, would enhance rigor.

We have added a reference to a recent study on changes in land cover for the same period: Calderon-Loor et al. 2021. High-resolution wall-to-wall land-cover mapping and land change assessment for Australia from 1985 to 2015. *Remote Sens. Environ.* 252

Page 7, line 13: The figure caption and associated text refers to times since fire, whereas the section heading is fire return interval. The inclusion in the text, figure and methods (page 21, lines 12 – 13) of time since fire is confusing and somewhat meaningless in this context and the narrative and figures should be based on actual intervals between fires.

We acknowledge that we inappropriately used different terminology interchangeably for terms that are not equivalent. That obscured the intent of the analysis and the rationale for our choices. We have now tightened the use of terminology and further explained, here and in the Methods section, what we did and why.

We agree that fire return interval, requiring pixels that have at least two fires, is the most robust method if data are available. Given the fire frequency of these forests is anywhere between 20 and more than 100 years, our 90-year record falls well short of capturing two fires for many fire regimes and forest types, let alone to be able to detect changes in trends over time, which is the focus of this study.

Alternatively, here we suggest years since the last wildfire (YSLF) as presented with our methods which provides:

- The most statistically robust approach as it allows a much larger pixel-year population to detect changes in trends over time. We were able to cover 155,384 km² of the 324,873 km² identified as forests in Australia covering all major forest types (see Methods: Forest extent and types).
- The YSLF gives the current state of the wildfire fire regime, whereas mean fire return interval lags one fire event behind. This way we provide a quantity that reflects the state all the way to the 2019 fire season.
- The initialization of the YSLF database is easy and enables us to use the information of both “fire” and “no-fire” for all pixels that burned at least once during the 90-year record. We have provided more detailed information on how we initialized the dataset in Methods: Years since Last Fire.

Page 8, line 6: McArthur’s FFDI indicates weather conditions across the ‘Danger’ spectrum, not just dangerous weather conditions.

We changed “weather conditions across the danger spectrum” to “dangerous weather conditions”.

Page 8, line 8: McArthur’s FFDI incorporates a drought index and a ‘Drought Factor’, not a fuel moisture deficit.

We have revised the wording and use Drought Factor as suggested.

Page 13, lines 9-10: There is another hypothesis regarding climate change effects on fuel loads, being that fuel loads are declining due to decreased vegetative productivity associated with drier conditions.

We agree with the reviewer, and we believe that this is likely the case in some regions based on the model results which show small decline in fine fuels. However, our coarse modelling shows that aggregated across all forests, the potential CO₂ fertilization effect is still winning over the negative effects of increased temperature and declining rainfall for most fuel types. We have revised the text to make this possible link to dryness clear:

“The simulation shows a small positive trend of potential aboveground biomass and coarse woody debris (Fig. 8a,b; Supplementary Table 4), but a declining trend of fine litter associated with declining canopy leaf biomass over Australian forests (Fig. 8b). This declining trend could be associated with the reduction in rainfall and soil moisture of the past four decades (Fig. 5e,g).”

‘Figure 4’ appears twice and Figure 5 caption is missing.

Thanks, it was a misnumbering of the figures. Corrected.

Page 14, line 2: 'prescribed burnings' should be written as 'prescribed burning'. Same for page 19, line 19.

Corrected throughout the ms.

Figure 4, the second one: There does not appear to be any orange-coded data indicated. If it is too limited in area to be visible in the figure then that should be indicated in the figure caption.

Yes, the amount of prescribed burning is so small in the ACT that it cannot be seen in the figure. We have now merged NSW with ACT in one single color. The merge also makes geographical sense.

Page 15, lines 6–10: Overall, if fine litter is projected to decline (Fig 6b), it is largely irrelevant for this study if the CO₂ effect in isolation is to increase aboveground biomass.

We have now rewritten some parts of the biomass section and we no longer make the CO₂ fertilization effect the main focus, but the combined effect of both climate change and CO₂. We agree that the net result of all drivers is what is most important. We only discuss individual drivers for their value in understanding the underlying causes of change.

Page 16, line 15 to page 17, line 2: A paragraph of the conclusion is effectively repeated.

Second paragraph deleted.

Some hyperlinks in the document are broken, e.g. Page 20, lines 30–31.

All links in the ms. now working.

Page 21, lines 2–3: The sentence does not make sense.

Fixed.

Page 21, lines 22–30: The figures and analyses should be limited to realized fire intervals, without including any aspect of synthetic fuel age. Given the clarification on page 22, line 2 then the method should just state the analysis is based on observations, without invoking the synthetic estimates of fuel age in the manuscript.

We have provided additional information in the Methods section explaining what we did. The initialization method we used allowed us to make fuller use of the observations in the State and Territory agency datasets. Specifically, the initialization method applies a probability distribution based on the 90-year data for each pixel (for pixels that have burned at least once in the record), in order to estimate when the last fire prior to 1930 might have occurred.

This approach allows us to use every single pixel-year of data from 1930 onwards, either as a fire or no-fire observation. For example, if a particular pixel first burned in 1980 and then a second time in 2006, by only using the second fire to determine a fire return interval, we would lose 50 years of no-fire observations in the record (1930-1979).

Because we only provide YSLF results for the last four decades, the mean values of those decades are little affected by the initialization, and we have been able to use a much larger pixel-year population on which to base the calculation. That is, by the time we reach the last decades of the fire record, and we calculate decadal means, the data feeding into the calculation are mostly real observations.

This approach is an attempt to overcome the limitations of the 90-year long dataset to study changes in trends over time, which is still too short given the long fire return intervals of many forest-types.

Page 23, lines 19–22: Again, the reliance on synthetic data is not convincing for this application.

We hope our response in the previous comment explains the rationale for the choice, and the fact that the initialization method has a small effect on the YSLF estimates for the last four decades, while increasing the statistical power to detect changes over time and larger forest area.

Table 1 – S1: The table needs to be carefully checked. There are several repeated pairs of dependent and independent variables (e.g. numbers 7 and 9; numbers 8 and 10).

We have checked, fixed and expanded the table. Note that some of the regressions are with and without the 2019 fire year to demonstrate that positive trends were already statistically significant before the extraordinary 2019 fire year. We have also expanded the regression analysis with new text in the section “Burned area and fire weather” with all parameters of the regressions in Supplementary Table 3.

REVIEWER COMMENTS

Reviewer #1 (Remarks to the Author):

This is the third time I have reviewed this manuscript. My main concern throughout the reviews has been the disconnect between the fuel load section and the fire weather variable analysis. I appreciate the authors response in not feeling that can make conclusions about the fuel parts and the fire weather parts however, they have put both topics in the same paper so it is logical that that need to be brought together otherwise they should separate these into two different papers. Nevertheless, to improve the manuscript, I believe the authors should add more details in the introduction on the fuel load component – what have other studies found? Why look at fuel hazard reduction burns? etc. This ensures the reader knows that fuel load is a key part of the analysis and understands the context as to why it has been added and what the authors are testing. The results section also needs to be strengthened to clarify why this is being analyzed, although this may be improved if added to the introduction. In contrary to these comments, I believe the fuel load section in the discussion and conclusion is now well integrated.

Otherwise I think the authors have done a good job of accommodating the requests of the reviewers and I believe some of the findings in this manuscript are novel and will be of great interest, particularly the area burned findings and related trends and figures. This will be of value to other researchers when exploring fire activity changes in relation to climate change and for fire/land managers in planning and adapting to a changing landscape.

Specific comments

Abstract – does not mention fuel load, again reflect the disconnect of that section to the remainder of the paper

The Introduction - does not adequately refer to existing literature on the key themes of the manuscript especially around fuel condition and availability.

Lines 21-23: The authors should add what they do with these data (i.e. analyse trends and variability) rather than just stating: "we use nine wildfire risk factors and indices....".

Lines 16-17: I do not think this is the correct reference: "and increasing firefighting capacity in Australia²⁸." If that is the correct reference then it is not a very strong reference to be arguing that fire fighting capacity has increase across Australia. AFAC or RCNNDA may have a better reference to use.

Lines 19-22: To better connect this section and to ensure readers understand why fuel hazard reduction burn trends are analysed there needs to be some context written here and in the introduction. What are fuel hazard reduction burns for, what is the hypothesis for analyzing the trend? Same for biomass production and fuel loads in relation to CO₂ and climate change. What have other studies found in relation to changes in fuel hazard reduction burn trends and biomass production?

Reviewer #2 (Remarks to the Author):

The authors have satisfactorily revised the manuscript in line with my prior comments. My condolences to the authors for the passing of co-author Vanessa Haverd.

A few minor points are:

1. Line 323 "32 year period (1988-2018)" – shouldn't this be changed to 32 (1988-2019) now? 2019 was added, yes?

Given the authors' reply "Given the time passed since the last review, we have now been able to acquire the AVHRR data for the 2019 fire season and we no longer need to complete the AVHRR series with MODIS for the 2019 season. We have permeated this change to all figures and trend analyses in the manuscript."

2. Figure 6 – was 2019 burned area from AVHRR added? If so update caption accordingly. If not, make sure this is the revised figure.

3. Supplementary figure 6 – same question

4. Supplementary table 1 – what is line 5? Explain in caption or remove.

5. Supplementary table 1 – should lines 12 and 13 have corresponding 2019 lines?

NCOMMS-20-29739B

REVIEWER COMMENTS

Reviewer #1 (Remarks to the Author):

This is the third time I have reviewed this manuscript. My main concern throughout the reviews has been the disconnect between the fuel load section and the fire weather variable analysis. I appreciate the authors response in not feeling that can make conclusions about the fuel parts and the fire weather parts however, they have put both topics in the same paper so it is logical that that need to be brought together otherwise they should separate these into two different papers. Nevertheless, to improve the manuscript, I believe the authors should add more details in the introduction on the fuel load component – what have other studies found? Why look at fuel hazard reduction burns? etc. This ensures the reader knows that fuel load is a key part of the analysis and understands the context as to why it has been added and what the authors are testing. The results section also needs to be strengthened to clarify why this is being analyzed, although this may be improved if added to the introduction. In contrary to these comments, I believe the fuel load section in the discussion and conclusion is now well integrated.

Thank you for your suggestions, we responded with what we have specific done to address the suggestions in the “Specific Comments” section below.

Otherwise I think the authors have done a good job of accommodating the requests of the reviewers and I believe some of the findings in this manuscript are novel and will be of great interest, particularly the area burned findings and related trends and figures. This will be of value to other researchers when exploring fire activity changes in relation to climate change and for fire/land managers in planning and adapting to a changing landscape.

We thank you for appreciating the effort we put to address your previous comments.

Specific comments

Abstract – does not mention fuel load, again reflect the disconnect of that section to the remainder of the paper.

Now, we specifically mention “fuel loads” in the list of variables that are part of this study:

“Through changes in the climate, anthropogenic climate change has the potential to alter fire dynamics. Here we compile the longest consistent satellite (19 and 32 years) and ground-based (90 years) burned area datasets, climate and weather observations, and **simulated fuel loads for Australian forests**”

The Introduction - does not adequately refer to existing literature on the key themes of the manuscript especially around fuel condition and availability.

We have added the following paragraph in the introduction with nine references to publish work on various aspects of the role of fuels and prescribed burning on fire risk:

“Fuel loads and trends, as effected by climate, human activity and time since the last disturbance, also play a role in determining fire risk^{23,24}. This link is a central motivation for using prescribed burning to reduce fuel availability^{25,26}, which in Australia is managed through changes in the frequency of prescribed burns²⁷. Although there is some debate on their value to reduce fire risk²⁸, particularly during extreme fire weather conditions^{2,25,29}, fuel loads and their distribution and structure are key determinants of fire spread, intensity and severity⁷. “

Lines 21-23: The authors should add what they do with these data (i.e. analyse trends and variability) rather than just stating: “we use nine wildfire risk factors and indices....”.

We have replaced “use nine...” with “analyse trends of nine...”

Lines 16-17: I do not think this is the correct reference: “and increasing firefighting capacity in Australia²⁸.” If that is the correct reference then it is not a very strong reference to be arguing that fire fighting capacity has increase across Australia. AFAC or RCNNDA may have a better reference to use.

We have added an additional reference and also made clear that we are not referring to the entire Australia but for bushfire firefighting largely in the Southeast states of the country.

ENRC. *Inquiry into the Impact of Public Land Management Practices on Bushfires in Victoria*. (2008).

Lines 19-22: To better connect this section and to ensure readers understand why fuel hazard reduction burn trends are analysed there needs to be some context written here and in the introduction. What are fuel hazard reduction burns for, what is the hypothesis for analyzing the trend? Same for biomass production and fuel loads in relation to CO₂ and climate change. What have other studies found in relation to changes in fuel hazard reduction burn trends and biomass production?

We have added a new paragraph in the introduction about the role of fuels and prescribed burning as shown above.

As per the paragraph in the fuels section (lines 19-22), and the lack of explanation on why we are analysing the datasets, we have added the following:

“Here we hypothesize that changes in trends leading to reduced fuel loads will reduce fire risk and burned area, while the opposite will be true for changes leading to increase fuel loads.”

Reviewer #2 (Remarks to the Author):

The authors have satisfactorily revised the manuscript in line with my prior comments. My condolences to the authors for the passing of co-author Vanessa Haverd.

A few minor points are:

1. Line 323 “32 year period (1988-2018)” – shouldn’t this be changed to 32 (1988-2019) now? 2019 was added, yes?

The regressions are based on 1988-2018 in purpose because we used the regressions to predict the 2019 burned area; so, we excluded 2019 with this purpose. What we want to show here is that what happened in 2019 could have been predicted based on what had already happened over the previous 32 years.

Given the authors’ reply “Given the time passed since the last review, we have now been able to acquire the AVHRR data for the 2019 fire season and we no longer need to complete the AVHRR series with MODIS for the 2019 season. We have permeated this change to all figures and trend analyses in the manuscript.”

2. Figure 6 – was 2019 burned area from AVHRR added? If so update caption accordingly. If not, make sure this is the revised figure.

Thanks, corrected, all data is from AVHRR.

3. Supplementary figure 6 – same question

The same as in 1)

4. Supplementary table 1 – what is line 5? Explain in caption or remove.

We have added the levels that FDDI corresponds to very high (≥ 25) and severe (≥ 50).

5. Supplementary table 1 – should lines 12 and 13 have corresponding 2019 lines?

These are the two equations we used to predict the burned area for the 2019 fire season, and therefore they need to stop in 2018.

REVIEWERS' COMMENTS

Reviewer #1 (Remarks to the Author):

I am satisfied that the authors have addressed all of my previous comments. I look forward to seeing this important work published.

I send my sincere condolences to the authors for the loss of their colleague Vanessa Haverd.